# Emergence of fractal geometries in the evolution of a metabolic enzyme

Franziska L. Sendker[1], Yat Kei Lo[2], Thomas Heimerl[2], Stefan Bohn[3], Louise J. Persson[4], Christopher-Nils Mais[5], Wiktoria Sadowska[6,7], Nicole Paczia[8], Eva Nußbaum[9], María del Carmen Sánchez Olmos[10], Karl Forchhammer[9], Daniel Schindler[2,10], Tobias J. Erb[2,8,11], Justin L. P. Benesch[6,7], Erik G. Marklund[4], Gert Bange[2,5,12], Jan M. Schuller[2,5 ✉] & Georg K. A. Hochberg[1,2,5 ✉]

Fractals are patterns that are self-similar across multiple length-scales[1]. Macroscopic fractals are common in nature[2–4]; however, so far, molecular assembly into fractals is restricted to synthetic systems[5–12]. Here we report the discovery of a natural protein, citrate synthase from the cyanobacterium *Synechococcus elongatus*, which self-assembles into Sierpiński triangles. Using cryo-electron microscopy, we reveal how the fractal assembles from a hexameric building block. Although different stimuli modulate the formation of fractal complexes and these complexes can regulate the enzymatic activity of citrate synthase in vitro, the fractal may not serve a physiological function in vivo. We use ancestral sequence reconstruction to retrace how the citrate synthase fractal evolved from non-fractal precursors, and the results suggest it may have emerged as a harmless evolutionary accident. Our findings expand the space of possible protein complexes and demonstrate that intricate and regulatable assemblies can evolve in a single substitution.

Fractals are repeating patterns in which substructures at smaller scales resemble structures at larger scales[1]. These shapes are fascinating because they can be constructed using simple mathematical rules that result in structures of extraordinary complexity and beauty[1]. Fractals exist in nature in the branching patterns of plant leaf veins, in coastal lines and in river systems[2–4]. Most natural fractals are irregular. That is, their structures at different scales do not exactly match. Instead, their self-similarity emerges from similar levels of detail in structures formed at different scales. The rare examples of regular fractals in nature, such as the repeating structures in Romanesco broccoli, have been intensely studied to understand the underlying mechanisms that produce exact self-similarity[13].

All known regular fractals in nature are made by living organisms and exist at the macroscopic scale. However, none have yet been discovered in nature at the molecular scale despite the extraordinary diversity of biomolecular assemblies known to science[14,15]. The reason for this may be that fractal construction algorithms are difficult to translate into molecular self-assembly. For example, the Sierpiński triangle, one of the best known regular fractals[16], can be created by triangular subdivision or through a stochastic 'chaos game' that relies on non-local rules[17], or by colouring in all elements of Pascal's triangle that have odd binomial coefficients. Conversely, self-assembly of biomolecules occurs through the sequential addition of subunits rather than by subdivision and relies on local contacts between protomers to coordinate assembly.

Synthetic designs have overcome these constraints and have built Sierpiński triangles out of small organic molecules. A key element of these designs is a molecular building block in which the structure traces out a 120° angle[5–8,12]. This angle matches that between subtriangles within the equilateral Sierpiński triangle and allows a single monomer to bridge between subtriangles in a manner that passivates their edges. This arrangement ensures that new triangles can only be attached on the tips of an existing triangle and thereby produces the large voids of the Sierpiński triangle. Other designs use metal coordination to enforce this angle[9–11]. All designer fractals require special surfaces, precise temperature control during assembly or finely tuned ratios of different precursors to produce Sierpiński fractals[5–12]. Such delicate assembly requirements are unlikely to be met in cells, which makes the possibility of natural versions of these fractals seem remote.

Here we report the discovery of a natural metabolic enzyme capable of forming Sierpiński triangles in dilute aqueous solution at room temperature. We determine the structure, assembly mechanism and its regulation of enzymatic activity in vitro and finally how it evolved from non-fractal precursors.

## A protein that forms Sierpiński triangles

Bacterial citrate synthase (CS) proteins are homo-oligomeric enzymes that can assemble into dimers and hexamers[18]. We discovered that

[1]Evolutionary Biochemistry Group, Max Planck Institute for Terrestrial Microbiology, Marburg, Germany. [2]Center for Synthetic Microbiology (SYNMIKRO), Philipps–University Marburg, Marburg, Germany. [3]Cryo-EM Platform and Institute of Structural Biology, Helmholtz Munich, Neuherberg, Germany. [4]Department of Chemistry – BMC, Uppsala University, Uppsala, Sweden. [5]Department of Chemistry, Philipps–University Marburg, Marburg, Germany. [6]Department of Chemistry, University of Oxford, Oxford, UK. [7]Kavli Institute for Nanoscience Discovery, Oxford, UK. [8]Department of Biochemistry and Synthetic Metabolism, Max Planck Institute for Terrestrial Microbiology, Marburg, Germany. [9]Interfaculty Institute of Microbiology and Infection Medicine, Organismic Interactions Department, Cluster of Excellence 'Controlling Microbes to Fight Infections', Tübingen University, Tübingen, Germany. [10]MaxGENESYS Biofoundry, Max Planck Institute for Terrestrial Microbiology, Marburg, Germany. [11]Department of Biology, Philipps–University Marburg, Marburg, Germany. [12]Max Planck Fellow Group Molecular Physiology of Microbes, Max Planck Institute for Terrestrial Microbiology, Marburg, Germany. ✉e-mail: jan.schuller@synmikro.uni-marburg.de; georg.hochberg@mpi-marburg.mpg.de

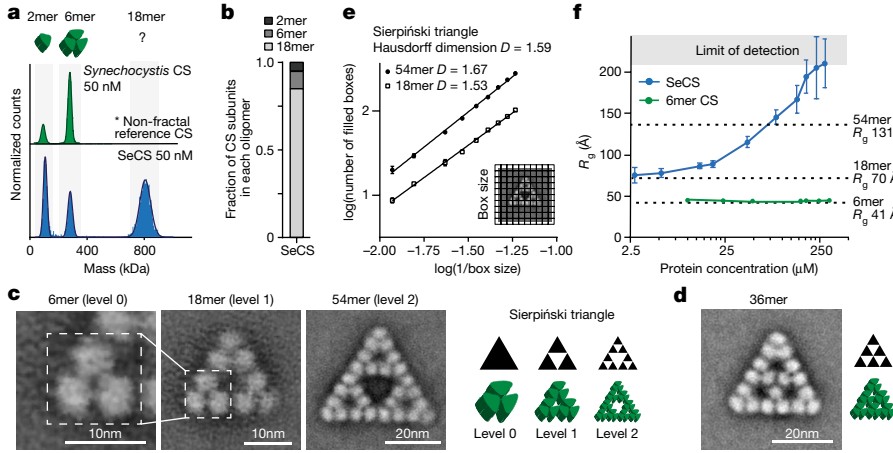

**Fig. 1 | CS of *S. elongatus* PCC 7942 assembles into Sierpiński triangles.**
**a**, Distribution of oligomeric protein complexes of purified CS from two cyanobacterial species, *S. elongatus* PCC 7942 (SeCS, monomer mass = 44.3 kDa) and *Synechocystis* sp. PCC 6803 (monomer mass = 45.9 kDa), measured by MP. Cartoons represent the assembly of known CS proteins. **b**, Distribution of SeCS subunits in the different oligomeric complexes corresponding to the MP measurements in **a**. **c,d**, 2D class averages of purified SeCS recorded by negative-stained EM. The 6mer complexes did not produce top-views (Extended Data Fig. 1b). Therefore, an isolated 6mer of the 18mer class average was used for representation. Schematics of the images are on the right. **e**, Box-counting quantification of the Hausdorff dimensions (*D*) using the class averages of the

18mers and 54mers. Data are presented as the mean values of three different grid positions, and error bars correspond to s.d. *D* was obtained from the slope of the regression line ($R^2_{54mer}$ = 0.996, $R^2_{18mer}$ = 0.997). **f**, $R_g$ values inferred from SAXS data for SeCS and a hexameric variant (SeCS L18Q; Fig. 5d) at varying protein concentrations. The experiment was conducted by starting at the highest concentration and then serially diluting the protein. Larger assemblies are therefore reversible. One sample for each concentration step was measured over ten frames. The data presented are the inferred $R_g$ values using Guinier approximation, and error bars correspond to the s.d. of fit values calculated from the covariance matrix (ScÅtter IV). Dashed lines indicate $R_g$ values calculated from the indicated structural models of the 6mer, 18mer and 54mer.

CS from the cyanobacterium *S. elongatus* PCC 7942 (SeCS) forms an unusual assembly. Mass photometry (MP) analyses of the purified enzyme at nanomolar concentrations revealed a complex comprising 18 CS subunits (Fig. 1a,b and Extended Data Fig. 1a).

We investigated the structure of these assemblies by negative-stain electron microscopy (EM) and observed that SeCS assembles into regular triangular complexes of different sizes (Fig. 1c,d and Extended Data Fig. 1b–f). The 18mer contains 9 discernible densities, each corresponding to a dimer. Three dimers are first arranged in a hexameric ring and three hexamers then connect into a triangle. This 18mer represented the main oligomeric species under MP conditions (>80% of all CS subunits at 50 nM; Fig. 1b). Rarely (on the order of 3–4% of particles; Methods), we observed even larger complexes comprising 36 or 54 CS subunits on the micrographs, which were recorded at a 9 times higher protein concentration than for MP (450 nM). The 54mer consisted of three 18mers arranged into an even larger triangle with a large void at its centre (Fig. 1c). The 6mer, 18mer and 54mer represent the zeroth, first and second order of the Sierpiński triangle, a well-known regular fractal geometry. The 36mer represents another kind of triangle, but shares the 6mer building block and the overall triangular shape (Fig. 1d). Additional regular assemblies that were sporadically observed also retained the triangular edges (Extended Data Fig. 1g).

To validate that the 18mer and the 54mer assemblies are fractal geometries, we approximated their Hausdorff dimension *D*. For non-fractal shapes, *D* takes on integer values (a square has 2, a cube has 3, and so on), whereas for fractals, it can be non-integer, with different fractals having their specific characteristic *D* values. We applied the box-counting method, which produced non-integer values that closely corresponded to the calculated fractal dimension of the Sierpiński triangle, considering also the limitations of box counting[19,20] (Fig. 1e; $D_{18mer}$ = 1.53 ± 0.02, $D_{54mer}$ = 1.67 ± 0.02, $D_{Sierpiński}$ = 1.59; mean ± s.d.). Mathematical fractals repeat themselves infinitely, so we explored whether our protein fractal could increase in size beyond 54 subunits. We used small-angle X-ray scattering (SAXS) measurements to assess the radius of gyration ($R_g$) in solution at a range of concentrations and compared our measured values to theoretical $R_g$ values calculated for structural models of the

6mer, 18mer, and 54mer (Supplementary Fig. 2). At concentrations above 100 μM (approximately 220 times higher than the concentrations used for negative-stain EM, and 2,000 times higher than for MP measurements), we measured $R_g$ values that exceeded the size of 54mers and rapidly reached the limit of detection (Fig. 1f). Although we cannot prove that the larger assemblies are Sierpiński triangles rather than some other type of assembly, these experiments indicate that the protein is capable of extended growth, as predicted for fractal assembly.

## Structural basis of Sierpiński assembly

To assemble a hexameric building block into a Sierpiński fractal, a new interface needs to be introduced between dimers. This interface has to satisfy two conditions. First, it has to connect hexamers along their 120° external angle to result in a triangle. Second, the interface has to be made between only two dimers, such that no more subunits can associate into the edges of the triangle (Fig. 2a). These criteria ensure that the edges of the triangles remain passivated and enables the large voids of the Sierpiński triangle to form. These demands are difficult to meet with protein–protein interfaces that are commonly found in homomers[21]. However, a 120° angle between hexamers can be achieved through the introduction of a three-fold symmetrical, head-to-tail C3 interface between dimers. But such an interface fails our second demand because it allows a third dimer to associate into it, which results in a triangular lattice and not a fractal. We could satisfy our second demand with a two-fold, head-to-head C2 interface between dimers, but this requires that hexamers no longer interact at the correct angle, which results in a hexagonal lattice (Fig. 2a).

To understand how the fractal forms assemble, we solved structures for the zeroth order (6mer, 3.1 Å), first order (18mer, 3.9 Å) and second order (54mer, 5.9 Å) of the protein Sierpiński triangle by cryo-EM (Fig. 2b, Extended Data Table 1 and Supplementary Figs. 3–5).

In the 18mer (level 1), CS dimers assemble into hexamers through a heterologous interface similar to known hexameric CS proteins. The 18mer is then formed through an additional contact between two dimers of adjacent hexamers. This interface has an unusual geometry,

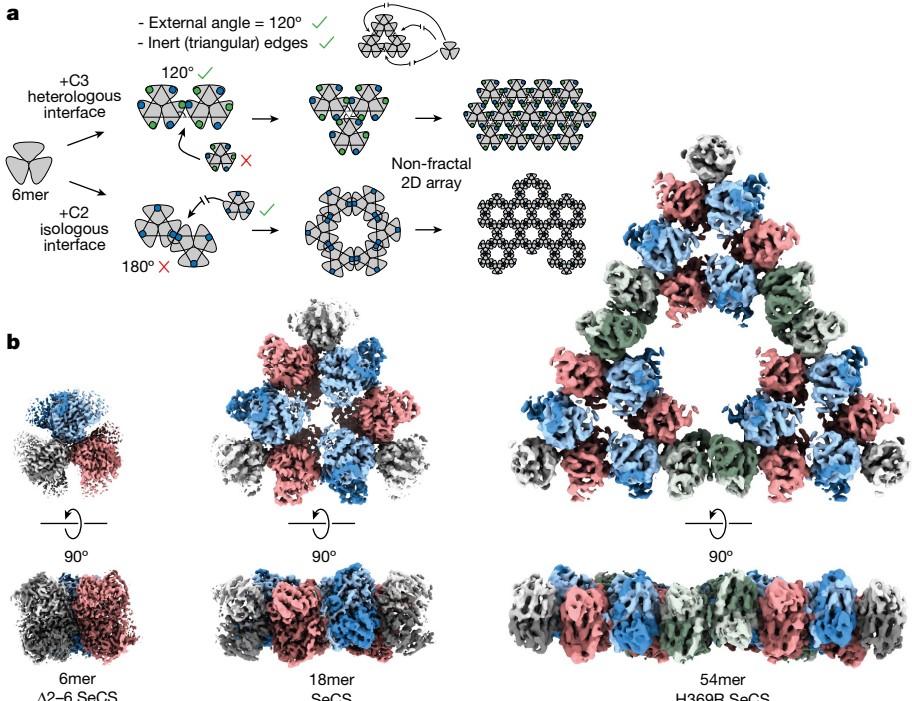

**Fig. 2 | Levels of fractal assembly. a**, Schematic representation of the requisites needed to produce a Sierpiński fractal from hexameric blocks and the symmetry-based constraints on oligomeric assembly. Green and blue dots represent active or open interfaces, respectively. **b**, Cryo-EM density maps of Sierpiński triangles of the zeroth, first and second fractal level. The 6mer (3.1 Å) was derived from the hexameric Δ2–6 SeCS variant. The 18mer (3.9 Å) was derived from the wild-type (WT) SeCS. The 54mer (5.9 Å) was derived from the pH-stabilized variant H369R SeCS.

whereby only one monomer from each dimer participates in the interaction. This is uncommon for interfaces that create dimers of dimers, in which normally all four chains participate equally in the interface[14]. The two participating monomers form an interface between the amino terminus of one monomer and the carboxy terminus of the other and vice versa (Fig. 3a and Extended Data Fig. 2a). An interaction between the residue E6 from one monomer and H369 of the other establishes an important contact, and mutating either E6 or H369 to alanine or deleting amino acids 2–6 (Δ2–6 SeCS) abolishes the formation of fractals (Extended Data Fig. 2b). In the two non-participating monomers, the termini face outwards. They cannot engage in the same interaction with another dimer without causing a steric clash (Extended Data Fig. 2c) and thereby passivate the edge of the triangular assembly. In the 18mer, the non-equivalence between chains allows the interaction to form at the correct angle and ensures that no more dimers can associate into it. This geometry further proves that 18mers are not merely a substructure of an otherwise ordinary triangular lattice but actually represent the first order of a Sierpiński triangle.

To understand how this interface forms, we compared the geometry of a variant that could only assemble into hexamers (Δ2–6 SeCS; Fig. 2b and Extended Data Fig. 2b) to that of a hexamer within the 18mer. We extracted a hexamer from the 18mer structure, aligned its free corner dimer onto one dimer of Δ2–6 SeCS and then compared the positions of the two unaligned dimers between the structures (Fig. 3b). This analysis revealed that the two dimers engaging in fractal connections undergo a small clockwise rotation relative to their conformation within a free hexamer, which subtly breaks the D3 symmetry of the hexameric building block within the fractal (Fig. 3b).

We next sought to understand how 18mers then assemble into 54-mers. We could not obtain a structure of the 54mer from wild-type SeCS because the protein tended to aggregate on the grids. Together with a strong tendency for preferential orientation, this resulted in too few top views of 54mers to solve a structure. Instead, we solved this structure for a point mutant (H369R), which we generated in the course of a structure–function analysis (described further below). This protein also forms 18mers at levels that are indistinguishable from wild-type SeCS at 50 nM (Extended Data Fig. 2b). Moreover, it is capable of forming the larger assemblies. It produced less aggregated cryo-EM grids, which enabled us to solve a 5.9 Å structure of the 54mer.

This structure first revealed that edges are passivated along its outward facing edges as well as those facing its large internal void, which did not allow hexamers to attach there without introducing steric clashes (Extended Data Fig. 2e,f). It also revealed that the interaction that holds 18mers into the 54mer uses the same surface as the interaction that holds 6mers into the 18mers. However, the dihedral angle between dimers across the interaction is different for the two types of connection within the 54mer (Fig. 3c). That is, the interaction between 6mers is made at 60°, which is identical to the angle at which it is made in a free 18mer. By contrast, the angle between 18mers within the 54mers is 34°. This different angle is necessary because the dihedral angle of interaction between hexamers is too large across the length of an 18mer to close the triangle in a 54mer (Extended Data Fig. 2g). For even larger assemblies, this angle would have to shrink further because it is amplified over even larger distances. The large assemblies we observed in our negative-stain EM experiments imply that small dihedral angles (which would be necessary to close large triangles) are possible. However, they may be less energetically favourable, which may explain why first-order fractals form readily at low nanomolar concentrations, whereas larger assemblies only occur at micromolar concentrations.

The 36mers and other complexes we sporadically observed seemed to partially violate the assembly rules we discovered in 18mers and 54mers. These complexes more closely resemble a two-dimensional (2D) crystal lattice, but with sharp triangular edges (Fig. 1c and Extended Data Fig. 1g). The main difference that distinguishes 36mers from the fractal 18mers and 54mers is a three-way junction of dimers in the centre of the 36mer. The interaction we discovered in 18mers cannot

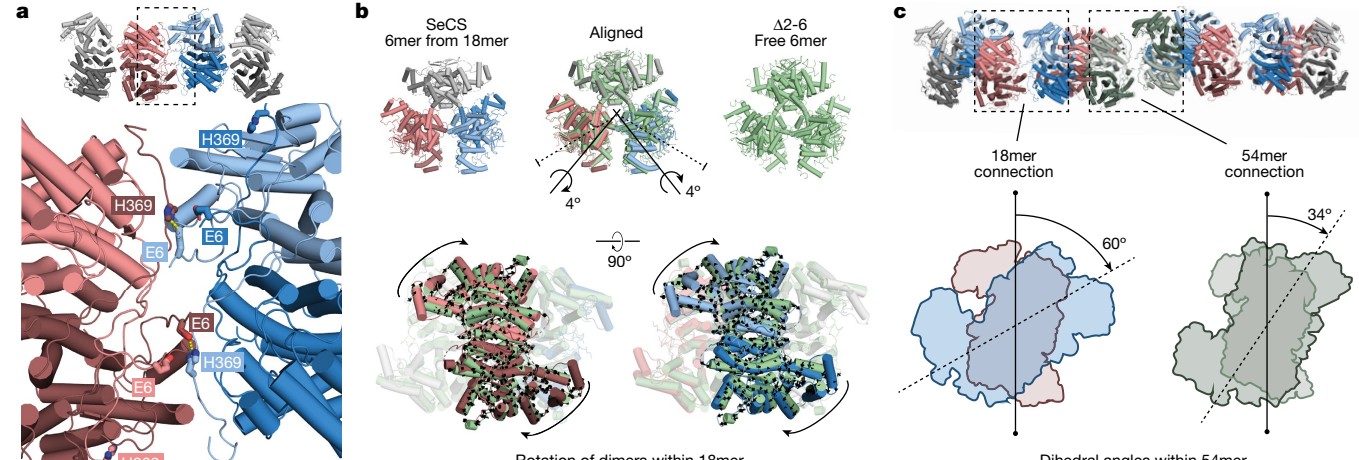

**Fig. 3 | The molecular basis of fractal assembly. a**, Close-up of the interface that produces the 18mer. Two out of the four adjacent protomers are making the connection (dark red and light blue). **b**, Structural alignment of a free Δ2–6 SeCS hexamer with an extracted hexamer from the 18mer structure. Black arrows indicate the molecular displacement in the 18mer, which corresponds to a rotation of the connecting dimers around an internal axis (black axis). The symmetry axes of the dimer interfaces are shown as dotted lines for reference. **c**, Dihedral angles between dimers interacting across the fractal interface are depicted for the connection within and between 18mers in a 54mer. Dimers are shown as blue and red, and green and olive outlines, respectively.

support such a three-fold association (Extended Data Fig. 3a). This result suggests one of two possibilities. One, this three-way junction is not a fully formed interface and the subunits are merely in close proximity. The other possibility is that this three-way junction is formed by a distinct C3 interface between dimers that we do not observe in our 18mer, which would imply that SeCS can also form regular triangular lattices (Extended Data Fig. 3a). Our 2D class average of the 36mer was not of sufficient resolution to distinguish between these possibilities, but two lines of evidence argue against an additional C3 interface. First, a mutation that ablates the fractal interface we see in the 18mer through a steric clash results in only hexamers and no larger stoichiometries (Fig. 1f). If an additional C3 interface exists, it must therefore overlap with the interface we see in 18mers. Second, we created a mutant that significantly weakened the interface to form hexameric subcomplexes but leaves the residues that form the fractal interaction intact (D147A). If an additional C3 interface is present, this variant should form hexamers and tetramers through the fractal interaction. We observed mostly tetramers with this mutant, with only a small fraction of hexamers and larger complexes (Extended Data Fig. 3b–i). This result implies that no strong additional C3 interface is present, which is also consistent with the fact that we see roughly similar numbers of 36mers and 54mers, despite the much larger subunit number of the latter (the number of particles observed in negative-stain EM was 186 for 36mers and 200 for 54mers).

Together, these observations reveal the larger principle at work from which Sierpiński triangles emerge. All assemblies we observed seemed to minimize the number of unsatisfied fractal interfaces. At stoichiometries for which Sierpiński triangles can be built (18, 54, 162, and so on), they are always the most efficient way to achieve this, leaving only 3 dimers at the corners of the triangle unsatisfied (Extended Data Fig. 4a). At intermediate stoichiometries, the protein apparently populates non-fractal but still triangular assemblies, which leave more dimers partially unsatisfied in the interior of the triangle (although we note that six 36mers could be arranged into a different type of fractal that is based on Pascal's triangle; Extended Data Fig. 4b). What unites all these assemblies is their distinct triangular shape, which also distinguishes them from the few other natural proteins that form 2D lattices[22]. Notably, SeCS achieves these assemblies without the help of metal coordination or symmetrical surfaces to assemble on, which synthetic Sierpiński triangles usually require. Instead, this ability emerges from the flexibility of individual protomers, breaking local symmetries as they assemble into higher-order structures. Conformational flexibility is a feature of many proteins[23], and there are no other obvious properties that make the hexameric building blocks of SeCS particularly suited to constructing Sierpiński triangles when compared with other dihedral homomeric protein complexes. Fractals and other non-crystalline 2D assemblies may therefore be constructed using a variety of other proteins and common assemblies as building blocks.

## Function of Sierpiński assemblies

We next investigated whether the assembly into 18mers, which were the dominant oligomeric species at physiological protein concentrations, has functional consequences for the enzyme. CS catalyses the condensation of acetyl-CoA and oxaloacetate to citrate as the first step in the tricarboxylic acid cycle. Addition of either of the substrates or the reaction product resulted in the disassembly of the structure into hexamers (Fig. 4a), which implies that hexamers are the catalytically active stoichiometry. We tested this idea in two ways. We first measured enzyme kinetics of the wild type and a variant that is incapable of forming fractals (SeCS L18Q). Under saturating substrate conditions, which completely disrupt fractal assembly, the kinetic parameters of both variants were almost identical (Fig. 4b and Extended Data Fig. 5a). Under non-saturating conditions, in which part of the 18mers remain, the wild-type SeCS was only half as active as the hexameric variant (Fig. 4b). Second, we constructed a mutant (cys4) that covalently stabilizes the fractal interface through a disulfide bridge. For this variant, a proportion of the fractal complexes remained at saturated substrate concentrations (Extended Data Fig. 5b). The $k_{cat}$ of cys4 was reduced compared with wild-type SeCS. This decline in activity was reversible after addition of reductant, which breaks the disulfide bridges (Extended Data Fig. 5c). These results imply that assembly into fractal complexes significantly reduces catalytic activity.

To understand why 18mers are less active, we solved a crystal structure of a citrate-bound hexamer at 2.7 Å (Extended Data Table 2 and Extended Data Fig. 6a). Comparing this structure to the one of a free hexamer (Δ2–6 SeCS) revealed that the CS dimers within the hexamer undergo an anticlockwise rotation after binding to citrate (Fig. 4c). This rigid-body rotation is well known from other CS enzymes[24,25]. It takes place following substrate binding and pushes CS from an open conformation (which corresponds to the conformation in the Δ2–6 SeCS structure) into a closed form in which catalysis takes place[26].

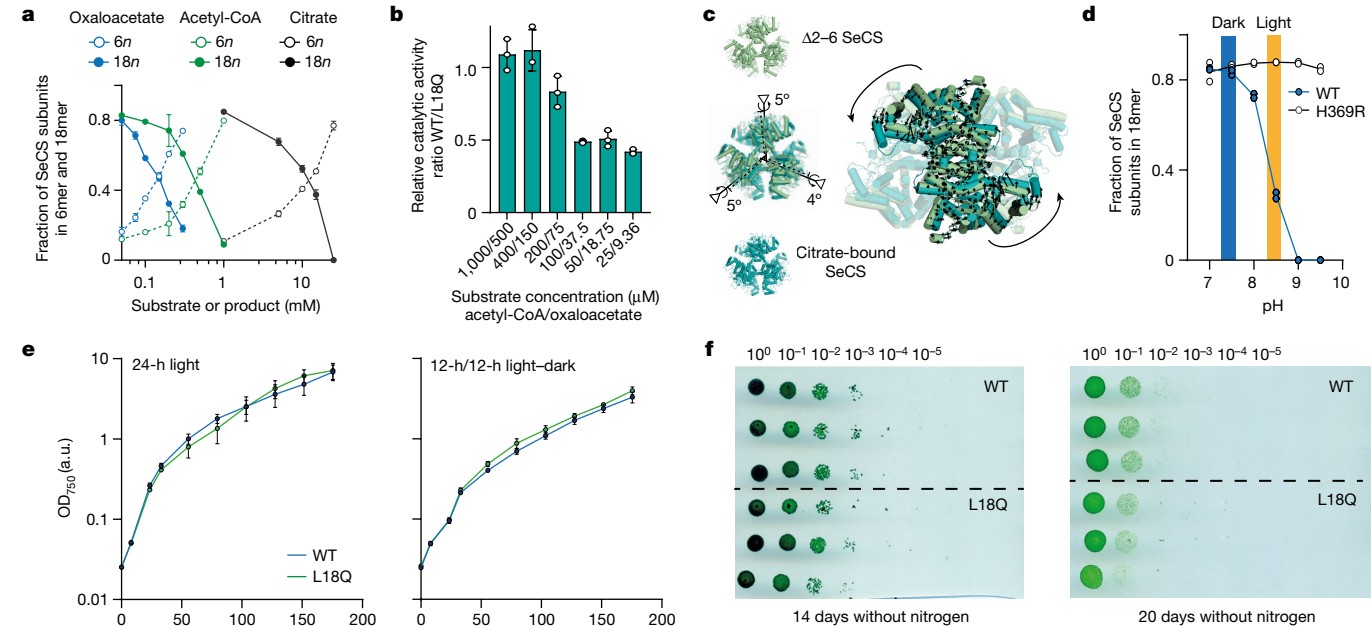

**Fig. 4 | Fractal formation affects catalysis in vitro but not fitness in vivo.**
**a**, Fraction of SeCS subunits in 6mers and 18mers at different concentrations of the two substrates acetyl-CoA and oxaloacetate and the product citrate quantified by MP. Data are presented as the mean values, and error bars indicate the s.d. $n = 3$. **b**, Ratio of turnover numbers (SeCS/L18Q, mean values, error bars indicate the s.d., $n = 3$) of SeCS and a hexameric variant (L18Q) at different substrate concentrations (see Extended Data Fig. 5 and Supplementary Table 1 for underlying data). **c**, Alignment of a citrate-bound structure with a Δ2−6 SeCS hexamer. Black arrows depict the molecular displacement from Δ2−6 SeCS into the citrate-bound state corresponding to a rotation around an internal axis within the dimers (black axes). The symmetry axes of the dimer interfaces are shown as dotted lines. Comparison of the citrate-bound conformation and the 18mer is shown in Extended Data Fig. 6b. **d**, MP quantification of SeCS subunits in 18mers at different pH values for the WT and the H369R variant. Protein concentration = 50 nM, $n = 2$. The intracellular pH range of *S. elongatus* in the dark and in light are indicated. Values taken from ref. 27. **e**, Growth curves of genetically modified *S. elongatus* strains harbouring either WT CS or hexameric L18Q CS at the original genetic locus and an additional antibiotic resistance cassette. Strains were cultivated either in full light (24 h) or light and dark cycles (12 h/12 h). Cultures were set up in three biological replicates, and data depict the mean values, error bars indicate the s.d. **f**, Survival of the genetically modified *S. elongatus* strains with either WT CS or the hexameric L18Q SeCS variant after nitrogen starvation for extended periods of time. Serial dilutions of three independent cultures are shown for each time point. Uncropped image in Supplementary Fig. 1b.

The rotation into the closed form is in the opposite direction to the clockwise rotation that dimers undergo to associate into the fractal. This result implies that fractal complexes have to perform a significantly larger conformational movement to induce substrate binding or catalysis, which probably imposes a higher energetic barrier that could explain the decrease in enzyme activity (Extended Data Fig. 6b,c).

We next asked whether this unusual assembly might have a function in *S. elongatus*, perhaps as a means to regulate the enzyme. We were led in this direction because fractal complexes are pH sensitive. Indeed, an increase in the pH from 7.5 to 9 led to complete disassembly of the structure into hexamers (Fig. 4d). This behaviour is driven by the residue H369 in the fractal interface, and changing it to arginine abolished the pH sensitivity without compromising its assembly into fractals (Fig. 4d and Extended Data Fig. 6d−g). Notably, *S. elongatus* undergoes a diurnal change in its intracellular pH that nearly matches the p$K_a$ of the fractal interface. During the day, its carbon concentration mechanism pushes the pH to 8.4 through the import of bicarbonate. At night, the pH returns to around 7.3 (ref. 27). Because fractal assembly reduces activity, the circadian pH shift could inhibit SeCS activity at night. This scenario seems plausible for two reasons: the intracellular substrate and product levels we observed were substantially below what would be necessary to disrupt 18mers (Extended Data Fig. 6h−i). Therefore, the assembly could reasonably form at night when the pH is sufficiently low. Second, CS is en route to 2-oxoglutarate, the only known precursor for nitrogen assimilation into glutamine and glutamate in *S. elongatus*, and several other enzymes in this pathway are known to be shut off at night[28]. To test this idea, we created genetically

modified strains carrying the wild-type CS or a mutant CS incapable of forming fractals at the native locus in *S. elongatus*. We quantified their growth under continuous light and under a 12-h day−night cycle, but found no differences in either condition (Fig. 4e). We also investigated whether the formation of fractals prevents the depletion of the tricarboxylic acid cycle owing to the synthesis of amino acids during nitrogen starvation. Such a regulation is described in the related cyanobacterium *Synechocystis* sp. PCC 6803 (ref. 29), but the regulators of this specific mechanism are not present in *S. elongatus*[30]. We therefore tested both genetically modified *S. elongatus* strains for recovery from nitrogen starvation. We again found no difference between the strains carrying the wild-type CS or non-fractal CS variant (Fig. 4f). Even though the assembly has many hallmarks of being regulatory (catalytic differences between stoichiometries, responsiveness to physiological conditions), it is apparently not important for fitness under our experimental conditions, although we cannot rule out that it might be under natural conditions.

## Evolution of the Sierpiński assembly

Based on these observations, we wondered whether this assembly could simply be an accident of history that is functionally inconsequential. We therefore sought to understand its evolutionary history. To that end, we first inferred a phylogenetic tree for CS proteins within Cyanobacteria, with CS proteins from marine Gammaproteobacteria as an outgroup (Fig. 5a and Supplementary Fig. 6). Based on this tree, we then characterized CS proteins from several relatives of *S. elongatus* (Extended

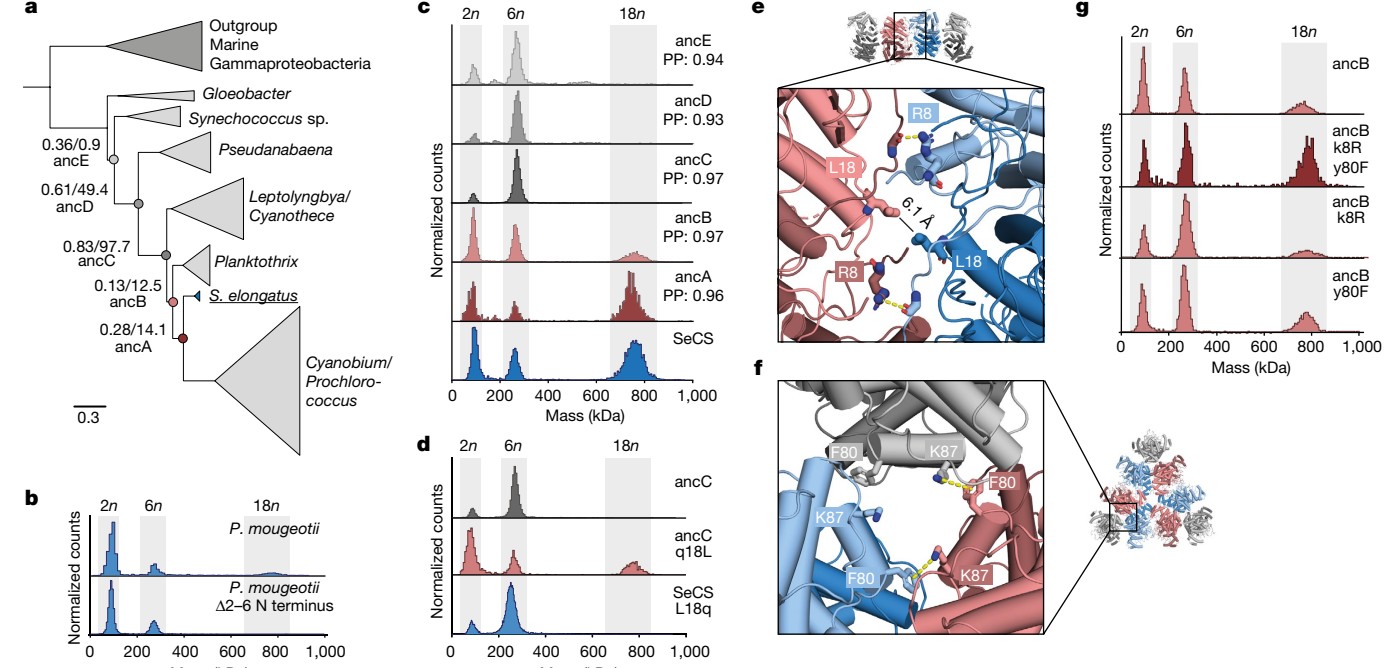

**Fig. 5 | Evolution of the SeCS fractal. a**, Phylogenetic tree of CS in Cyanobacteria (full phylogeny in Supplementary Fig. 6). Internal nodes that were resurrected by ancestral sequence reconstruction are labelled with branch support by Felsenstein's bootstrap and approximate likelihood test statistic. **b**, MP of purified extant CS from the *Planktothrix* clade and its N-terminal truncation. **c**, MP of purified ancestral CS. PP, average posterior probability of the maximum a posteriori state of the reconstructed sequence over all amino acids. **d,g**, MP of ancestral CS proteins with individual historical substitutions (lowercase and uppercase letters refer to ancestral and derived amino acids, respectively). **e,f**, Location of the substitutions within the structure of the SeCS 18mer.

Data Fig. 7a). We did not find 18mers in any of these relatives, except for *Planktothrix mougeotii*. Here we observed a very low abundance of 18mers that could be abolished by deleting the same N-terminal residues that abrogate assembly in SeCS (Fig. 5b). This result implies that 18mers, and by extension fractals, must have evolved along the lineage to *S. elongatus*, probably in the last common ancestor of *P. mougeotii* and *S. elongatus* and were then quickly lost again along the lineage to *Cyanobium/Prochlorococcus*.

To test this theory, we used ancestral sequence reconstruction to resurrect ancestral CS proteins at successive nodes leading from SeCS towards the root and characterized their assemblies by MP (Fig. 5c). The most recent of the ancestors (ancA) also formed 18mers at nanomolar concentrations and at abundances similar to SeCS. ancB, the immediate predecessor of ancA, also formed 18mers, but at a lower abundance, which indicated a weaker fractal interface. Results from SAXS analyses also verified that ancB can form larger assemblies at higher concentrations (Extended Data Fig. 7b). All subsequent ancestors (ancC–ancE) only formed hexamers. Together, these data suggest that relatively weak fractals evolved from hexamers between ancC and ancB, which then became stronger between ancB and ancA. Similar results were obtained when resurrecting alternative, less likely sequences for these ancestors (Extended Data Fig. 8a–e).

We then sought to identify which historical amino acid substitutions enabled fractal assembly. Notably, the side chains E6 and H369, which form the crucial and pH-sensitive contact of the fractal interface, are already present in ancestors that do not yet form fractals (Extended Data Fig. 8f,g). Only one substitution, q18L (lowercase and uppercase letters refer to ancestral and derived amino acids, respectively) occurred in the fractal interface within the interval between ancC and ancB, when fractals first evolved. Introducing only q18L into ancC was sufficient to trigger the formation of fractal complexes (Fig. 5d), including complexes larger than 18mers (Extended Data Fig. 8h). Reversing this substitution (L18q) in SeCS abolished fractal assembly (Fig. 5d). In

our 18mer structure, the L18 side chains from the two non-participating monomers are at the centre of the fractal interface (Fig. 5e and Extended Data Fig. 8i), but do not interact across the interface. q18L therefore probably removed a repulsive polar interaction from the interface that prevented fractals from forming in ancC. Introducing the substitution q18L into more ancient ancestors than ancC did not, however, produce fractals (Extended Data Fig. 8j). The historical window of opportunity in which fractals could evolve through just the q18L substitution was therefore very narrow.

We next searched for substitutions that strengthened the fractal interface along the interval between ancB and ancA. Only two conservative substitutions occurred at interfaces along this interval: k8R, which is located in the fractal interface and potentially allowed a more stable hydrogen bonding interaction with the backbone of the opposing monomer (Fig. 5e and Extended Data Fig. 8k), and y80F, which is located in the older interface connecting dimers into hexamers. In SeCS, our density analysis suggested that F80 engages in a cation–π interaction across the hexamer interface (Fig. 5f and Extended Data Fig. 8l). y80F may have affected the strength of this interaction[31], potentially making the dimer rotations that are necessary to associate into 18mers more favourable. Neither substitution alone led to more stable 18mers when introduced into ancB. Both substitutions together, however, produced a fraction of 18mers comparable with that of ancA (Fig. 5g). Conversely, reversing these historical substitutions in wild-type SeCS destabilized the 18mers (R8k, F80y; Extended Data Fig. 8m).

Together, our results show that building a stable fractal required a markedly small number of substitutions and no new strong contacts in the fractal interface. This result is consistent with previous studies showing that individual substitutions can substantially shift the occupancy of oligomeric states or induce supramolecular assembly[32–36]. This facile origin and almost immediate subsequent losses in two lineages (Extended Data Fig. 8b) make a non-adaptive origin at least plausible. Moreover, the $pK_a$ of the new interface was already primed to match the

physiological pH fluctuations as soon as the assembly emerged: introducing q18L into ancC (which, without this mutation, cannot make 18mers) resulted in an interface that dissociated over a similar pH range as in wild-type SeCS (Extended Data Fig. 8n). The apparently physiologically tuned p$K_a$ of the assembly is therefore either a molecular spandrel (a feature that initially evolved for reasons unrelated to its current use[37]) or simply a coincidence that looks deceptively adaptive. The large assemblies this protein can build are more clearly an evolutionary accident: as beautifully regular and complex as higher-order Sierpiński triangles are, they only appear at non-physiological concentrations and their size would be difficult to fit into the cytoplasm of *S. elongatus*. Even if the 18mer fulfils a useful function, the ability of the protein to make larger assemblies is almost certainly an accidental by-product of the unusual symmetry it happened to have evolved.

Based on our findings, we suspect that evolutionary transitions in self-assembly may be more common than structural databases make it seem. As techniques to characterize protein complexes improve, we may discover a menagerie of assembly types that come and go as proteins evolve. Perhaps only a small fraction ever become important to their organisms and persist. Many others will fade as quickly as they appeared. We can therefore only wonder about how many unique assemblies have evolved over the eons that never made it to the present for us to observe.

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

# Methods

## Molecular cloning

The gene encoding CS from *S. elongatus* PCC 7942 was amplified from genomic DNA by PCR (Q5 High-Fidelity 2× Master Mix, New England Biolabs) and introduced into the pLIC expression vector[38] by Gibson cloning (Gibson Assembly Master Mix, New England Biolabs). All other extant and ancestral CS sequences were obtained as gene fragments from Twist Bioscience and introduced into the same expression vector by Gibson cloning. All CS sequences were tagged with a C-terminal poly-histidine tag for purification (tag sequence: LE-HHHHHH-Stop). For single-site mutants and deletions of the CS sequences, KLD enzyme mix (New England Biolabs) was used. Mutagenesis primers were designed with NEBasechanger and used to PCR-amplify the vector encoding for the gene that was to be changed. Resulting PCR products were added to the KLD enzyme mix and subsequently transformed. All cloned genes were verified by Sanger sequencing (Microsynth) before use in experiments.

The DNA sequences of all purified proteins and NCBI identifiers of all extant sequences are presented in Supplementary Table 2.

## Protein purification

For heterologous overexpression, the vectors with the gene of interest were transformed into chemically competent *Escherichia coli* BL21 (DE3) cells. Transformed colonies were used to inoculate expression cultures (500 ml) made from LB medium supplemented with 12.5 g l$^{-1}$ lactose (Fisher Chemical). The cultures were incubated overnight at 30 °C and 200 r.p.m. Cells were collected by centrifugation (4,500$g$, 15 min, 4 °C), resuspended in buffer A (20 mM Tris, 300 mM NaCl and 20 mM imidazole, pH 8) and freshly supplemented with DNAse I (3 units μl$^{-1}$, Applichem). The cells were disrupted using a Microfluid-izer (Microfluidics) in 3 cycles at 15,000 psi and centrifuged to spin down cell debris and aggregates (30,000$g$, 30 min, 4 °C). The clarified lysate was loaded with a peristaltic pump (Hei-FLOW 06, Heidolph) on prepacked nickel-NTA columns (5 ml Nuvia IMAC Ni-Charged, Bio-Rad) that were pre-equilibrated with buffer A. The loaded column was first washed with buffer A for 7 column volumes and then with 10% (v/v) buffer B (20 mM Tris, 300 mM NaCl and 500 mM imidazole, pH 8) in buffer A for 7 column volumes. The bound protein was eluted with buffer B and either buffer-exchanged with PD-10 desalting columns (Cytiva) into PBS or 20 mM Tris, 200 mM NaCl, pH 7.5 or further puri-fied by size-exclusion chromatography (SEC). For SEC, the protein was injected on an ENrich SEC 650 column (Bio-Rad) with PBS as the running buffer using a NGC Chromatography System (Bio-Rad). The purity of the proteins was analysed by SDS–PAGE. After either buffer exchange or SEC, the purified proteins were flash-frozen with liquid nitrogen and stored at −20 °C before further use.

## Phylogenetic analysis and ancestral sequence reconstruction

Amino acid sequences of 84 CS genes from Cyanobacteria and marine Gammaproteobacteria as the outgroup were collected from the NCBI Reference Sequence database and aligned using MUSCLE (v.3.8.31)[39]. The maximum likelihood (ML) phylogeny was inferred from the mul-tiple sequence alignment (MSA) using raxML (v.8.2.10)[40]. The LG sub-stitution matrix[41] was used as determined by automatic best-fit model selection as well as fixed base frequencies and a gamma model of rate heterogeneity. The robustness of the ML tree topology was assessed by inferring 100 non-parametric bootstrap trees with raxML, from which Felsenstein's and transfer bootstrap values were derived using BOOSTER (https://booster.pasteur.fr). Using PhyML (3.0)[42], we also inferred approximate likelihood-ratio test[43] for branches to statistically evaluate branch support in the phylogeny.

Based on the CS tree and the MSA, ancestral sequences were inferred using the codeML package within PAML (v.4.9)[44]. To adjust for gaps and the different lengths of N termini in the CS sequences, their ancestral state was determined using parsimony inference in PAUP (4.0a) based on a binary version of the MSA (1 = amino acid, 0 = gap, no residue). The state assignment for each node in the tree (amino acid or gap) was then applied to the inferred ancestral sequences.

The initial reconstruction of the crucial amino acid substitution q18L was ambiguous in the ancestors ancB and ancA. We determined that this was the case because this L residue is present in the *Planktothrix* clade and *S. elongatus*, but not in the *Cyanobium/Prochlorococcus* clade. Therefore, the L residue was either gained once and then lost along the lineage to *Cyanobium/Prochlorococcus* or it was gained conver-gently twice in *Planktothrix* and *S. elongatus*. We therefore added the CS sequence from the cyanobacterium *Prochlorothrix hollandica* to the alignment, which has been stably inferred as a sister group to *S. elongatus* and the *Cyanobium/Prochlorococcus* clade by multiple studies[45–47]. This sequence was previously omitted from analysis as its position on the tree could not be inferred with high support. We manually added a branch to the tree, placing *P. hollandica* as a sister group to *S. elon-gatus* and the *Cyanobium/Prochlorococcus* clade. Branch lengths were reoptimized using raxML and the ancestral reconstruction repeated using PAML. The results gave high support to the substitution q18L being found in both ancB and ancA (Extended Data Fig. 8b) because *P. hollandica* also contains the L at position 18. This made the hypothesis of one gain and a subsequent loss in the *Cyanobium/Prochlorococcus* clade much more probable compared with three independent gains in *Planktothrix*, *P. hollandica* and *S. elongatus*.

## MP analysis

Measurements were performed on a OneMP mass photometer (Refeyn). Reusable silicone gaskets (CultureWellTM, CW-50R-1.0, 50-3 mm diam-eter × 1 mm depth) were set up on a cleaned microscopic cover slip (1.5 H, 24 × 60 mm, Carl Roth) and mounted on the stage of the mass photometer using immersion oil (IMMOIL-F30CC, Olympus). The gasket was filled with 19 μl buffer (PBS or 20 mM Tris, 200 mM NaCl pH 7.5) to focus the instrument. Then, 1 μl of prediluted protein solution (1 μM) was added to the buffer droplet and thoroughly mixed. The final concentration of the proteins during measurement was 50 nM unless stated otherwise. Data were acquired for 60 s at 100 frames per s using AcquireMP (Refeyn, v.1.2.1). The resulting movies were processed and analysed using DiscoverMP (Refeyn, v.2.5.0). The identified protein complexes with corresponding molecular weight were plotted as his-tograms, and the individual oligomeric state populations appeared as peaks that were fitted by a Gaussian curve (implemented in Discov-erMP). All complexes within the respective Gaussian curve were used to calculate the fraction of CS subunits in each oligomeric state. The instrument was calibrated at least once during each measuring session using either a commercial standard (NativeMark unstained protein standard, Thermo Fisher) or a homemade calibration standard of a protein with known sizes of complexes.

For substrate titrations, the prediluted protein sample (2 μM) was incubated for 10 min with the respective substrate concentration. The same substrate concentration was also included in the buffer in the gasket that was used for focusing. For each substrate concentration, three separate measurements were performed. For pH titrations, the protein sample was diluted into the buffer with the corresponding pH value (20 mM Tris, 200 mM NaCl pH 7–9.5). The dilution factor was at least 200, including predilution and final dilution in the gasket. For each pH value, two separate measurements were performed.

## Native mass spectrometry

The purified protein samples were buffer-exchanged into 200 mM ammonium acetate by using centrifugal filter devices (Amicon Ultra) and three successive rounds of concentration and dilution. The con-centration of protein was determined by UV absorbance (NanoDrop spectrophotometer, Thermo Fisher) and diluted into aliquots at appro-priate monomeric concentrations. Nanoelectrospray was carried

out in positive-ion mode on a Q Exactive UHMR mass spectrometer (Thermo Fisher), using gold-coated capillaries prepared in-house and the application of a modest backing pressure (about 0.5 mbar). Sulfur hexafluoride was introduced into the collision 'HCD' cell to improve transmission, and the instrument was operated at a resolution of 6,250 (at 200 $m/z$), with 'high detector optimization', and a trapping pressure in the HCD cell set to 4. The rest of the parameters were optimized for each sample, with the following ranges: capillary voltage of 1.2–1.5 kV; capillary temperature of 100–250 °C; in-source trapping from −15 to −150 V; injection times of 50–100 ms; and 1–10 microscans. Mass spectra were deconvolved using UniDec[48].

### Kinetic enzyme assays

For the CS kinetic assays, the colorimetric quantification of thiol groups was used based on 5,5′-dithiobis-(2-nitrobenzoic acid) (DTNB)[49,50]. The photospectrometric reactions were carried out in 50 mM Tris pH 7.5, 10 mM KCl, 0.1 mg ml$^{-1}$ DTNB and 25 nM protein concentration at 25 °C. To measure $K_m$ values, one substrate was saturated and added to the reaction mix (1 mM oxaloacetate or 0.5 mM acetyl-CoA). The other substrate was varied in concentration and added last to start the reaction. For kinetic measurements at non-saturating substrate concentrations, the protein was diluted only immediately before the reaction start to prevent the disassembly of complexes and added last to the reaction mix. Reaction progress was followed by measuring the appearance of 2-nitro-5-thiobenzoate at 412 nm (extinction coefficient of 14.150 M$^{-1}$ cm$^{-1}$) in a plate reader (Infinite M Nano+, Tecan) using Tecan i-control (v.3.9.1). Data analysis and determination of Michaelis–Menten kinetic parameters was done using GraphPad Prism (v.8.4.3). For the kinetic assays with the cys4 variant, the protein was dialysed in a buffer with a glutathione redox system to induce the formation of disulfide bonds of the cysteine residues (50 mM Na$_2$HPO$_4$, 150 mM NaCl, 1 mM glutathione and 0.5 mM glutathione disulfide, pH 8). After overnight dialysis, part of the protein sample was used for kinetic assays. The remainder was reduced by incubation with 10 mM dithiothreitol for 3 h at 4 °C and again used for kinetic assays. To exclude additional effects by the treatment itself, the WT SeCS was handled accordingly (dialysis in redox buffer and reduction with dithiothreitol) and measured kinetically for comparison.

### Box counting

To quantify fractal scaling, we used a fixed grid scan. The images of the class averages of the 18mer and 54mer assemblies were overlaid with a non-overlapping regular grid (Adobe Illustrator, v.24.0.2). The squares that were needed to fill out the structure were manually counted. This process was repeated for nine different box sizes of the grid (85–17 px). The entire procedure was replicated for three separate grid orientations for both assemblies. Linear regression was performed using GraphPad Prism (v.8.4.3).

### Cultivation of *S. elongatus* and sample preparation for metabolomics analysis

*S. elongatus* PCC 7942 was genetically modified to harbour variants of CS by homologous recombination as previously described[51]. The standard vector pSyn_6 (Thermo Fisher Scientific) was used as the backbone. A homology cassette was constructed by amplification and extraction of the CS gene and 1,000 bp of the neighbouring homologous regions by PCR from genomic DNA of WT *S. elongatus* PCC 7942. These were introduced into the pSyn_6 vector that included a spectinomycin-resistance gene to select for transformants. The respective sequence changes of the CS were introduced into this vector (L18Q) to create the corresponding homology cassette. The constructed homology cassettes (WT, L18Q) were transformed into WT *S. elongatus* PCC 7942 and plated on BG11 plates with 10 μg ml$^{-1}$ spectinomycin for selection. Transformants were re-streaked on fresh BG11 plates with spectinomycin, and resulting colonies were analysed

for successful integration through the extraction of genomic DNA. All strains were verified by PCR amplification of the introduced cassette (primers were designed to bind outside the introduced DNA region) and Sanger sequencing. All sequences of the homology cassettes are presented in Supplementary Table 3.

*S. elongatus* PCC 7942 cultures and genetically modified strains were grown in BG11 medium at 30 °C, 100 r.p.m., ambient CO$_2$ levels and alternating light conditions: 12 h of light (photon flux of 120 μmol m$^{-2}$ s$^{-1}$) and 12 h of darkness. Before the growth experiment, precultures were entrained for 5 days in the circadian conditions to synchronize cells. Then 3 main cultures (50 ml) were set up from 3 independent precultures and inoculated to an OD$_{750}$ of 0.025 or 0.05. Samples for metabolomics analysis were cultivated in specific flasks to facilitate the isolation of culture solution through a syringe valve, which led to slower growth behaviour compared with the standard flasks. The samples were taken at 6 different time points (days 3, 5 and 7) after a light and a dark period.

For recovery experiments under nitrogen deficiency, *S. elongatus* strains were grown in BG11 medium at full light to an OD$_{750}$ of 0.5 in triplicate. The cells were then shifted to medium without a nitrogen source. To do this, the cells were washed twice with BG11 without nitrate and then continuously cultivated in BG11 without nitrogen. The cells underwent chlorosis and fully bleached in the subsequent days. After 14 and 20 days, a serial dilution of the respective cultures was spotted on BG11 agar plates and incubated for 7 further days for recovery.

### Sample preparation for metabolomics analysis

The culture volume (1 ml) was taken from the shaking flask through a syringe and immediately quenched in 1 ml 70 % methanol that was precooled in a −80 °C freezer. The sample was mixed and centrifuged (10 min, −10 °C, 13,000$g$). The supernatant was removed and the pellet was stored at −80 °C until the endometabolome was extracted. At each time point, the cell number and size were measured for each culture using a Coulter counter (Multisizer 4e, Beckman Coulter). The respective biovolume for each cell pellet was then calculated and used to infer a normalized amount of extraction fluid for each sample (extraction fluid = 20,00 × biovolume). All steps of the metabolome extraction were performed on ice and with precooled (−20 °C) reagents. To extract the metabolites, the calculated amount of extraction fluid (50% (v/v) methanol, 50% (v/v) TE buffer pH 7.0) was added to the cell pellets together with the same amount of chloroform. The samples were vortexed and incubated for 2 h at 4 °C while shaking. The phases were then separated by centrifugation (10 min, −10 °C, 13,000$g$). The upper phase was extracted with a syringe and the same amount of chloroform added again. After mixing, the sample was centrifuged again (10 min, −10 °C, 13,000$g$) to get remove residual cell fragments and pigments. The upper phase was isolated, added to LC–MS vials and stored at −20 °C until analysis.

### Quantification of intracellular metabolites from *S. elongatus* by LC–MS/MS

Quantitative determination of acetyl-CoA and citrate was performed using LC–MS/MS. The chromatographic separation was performed on an Agilent Infinity II 1290 HPLC system (Agilent) using a Kinetex EVO C18 column (150 × 2.1 mm, 3 μm particle size, 100 Å pore size, Phenomenex) connected to a guard column of similar specificity (20 × 2.1 mm, 3 μm particle size, Phenomoenex). For acetyl-CoA, a constant flow rate of 0.25 ml min$^{-1}$ with mobile phase A being 50 mM ammonium acetate in water at a pH of 8.1 and phase B being 100% methanol at 25 °C was used. The injection volume was 1 μl. The mobile phase profile consisted of the following steps and linear gradients: 0–0.5 min constant at 5% B; 0.5–6.5 min from 5 to 80% B; 6.5–7.5 min constant at 80% B; 7.5–7.6 min from 80 to 5% B; and 7.6 to 10 min constant at 5% B. An Agilent 6470 mass spectrometer (Agilent) was used in positive mode with an electrospray ionization (ESI) source and the following conditions: ESI spray voltage

of 4,500 V; nozzle voltage of 1,500 V; sheath gas of 400 °C at 11 l min$^{-1}$; nebulizer pressure of 30 psi; and drying gas of 250 °C at 11 l min$^{-1}$. The target analyte was identified based on the two specific mass transitions (810.1 → 428 and 810.1 → 302.2) at a collision energy of 35 V and its retention time compared with standards.

For citrate, a constant flow rate of 0.2 ml min$^{-1}$ with mobile phase A being 0.1% formic acid in water and phase B being 0.1% formic acid methanol at 25 °C was used. The injection volume was 10 µl. The mobile phase profile consisted of the following steps and linear gradients: 0–5 min constant at 0% B; 5–6 min from 0 to 100% B; 6–8 min constant at 100% B; 8–8.1 min from 100 to 0% B; and 8.1 to 12 min constant at 0% B. An Agilent 6495 ion funnel mass spectrometer (Agilent) was used in negative mode with an ESI source and the following conditions: ESI spray voltage of 2,000 V; nozzle voltage of 500 V; sheath gas of 260 °C at 10 l min$^{-1}$; nebulizer pressure of 35 psi; and drying gas of 100 °C at 13 l min$^{-1}$. The target analyte was identified based on the two specific mass transitions (191 → 111.1 and 191 → 85.1) at a collision energy of 11 and 14 V and its retention time compared with standards.

Chromatograms were integrated using MassHunter software (Agilent). Absolute concentrations were calculated based on an external calibration curve prepared in sample matrix.

## Negative-stain EM

Carbon-coated copper grids (400 mesh) were hydrophilized by glow discharging (PELCO easiGlow, Ted Pella). Next, 5 µl of 450 nM protein suspensions were applied onto the hydrophilized grids and stained with 2% uranyl acetate after a short washing step with double-distilled H$_2$O. Samples were analysed using a JEOL JEM-2100 transmission electron microscope with an acceleration voltage of 120 kV. A 2k F214 FastScan CCD camera (TVIPS) was used for image acquisition. Alternatively, a JEOL JEM1400 TEM (operated at 80 kV) with a 4k TVIPS TemCam XF416 camera was used. For 2D class averaging, images were taken manually and processed with cisTEM[52]. The following number of particles were averaged: 1,491 particles for 18mers; 200 particles for 54mers; and 186 for 36mers. The 36mer and 54mer particles were isolated from an extended dataset, in which we specifically looked for larger assemblies. The exact percentage of complexes larger than 18mers was difficult to estimate because of very strong preferential orientation. Most particles seemed to have landed not on the face of the triangle but on one of its edges or even one of its tips (Extended Data Fig. 1). To obtain an estimate, another dataset of 150 micrographs without a bias towards larger assemblies was collected. All particles were manually counted for these micrographs and included the assemblies that were laying on their edge and appeared as rectangles. By measuring the edge length, we could assign them to be either a 36mers (30 nm) or 54mers (40 nm). The analysis revealed that under negative-stain TEM conditions (450 nM) approximately 92.8% of detected assemblies were identified as 18mers (1,773 particles), 3.5% as 36mers (66 particles) and 3.8% as 54mers (72 particles). Our estimate of the abundance should still be taken with care and by comparison with our SAXS data, which showed that large complexes only start being reasonably common above 25 µM protein concentration. For the H369R variant of SeCS, a protein concentration of 450 nM was used and 136 particles were averaged to produce the 2D class average of the 18mer shown in Extended Data Fig. 6d.

## Crystallography and structure determination

Crystallization was performed using the sitting-drop method at 20 °C in 250 nl drops (Crystal Gryphon, Art Robbins Instruments) consisting of equal parts of protein and precipitation solutions (Swissci 3 Lens Crystallisation Plate). Protein solutions of 250 µM were incubated with 5 mM acetyl-CoA for 10 min at room temperature to induce disassembly into hexamers. The crystallization condition was 0.1 M citrate pH 5.5, and 2.0 M ammonium sulfate. Before data collection, crystals were flash-frozen in liquid nitrogen using a cryo-solution that consisted of motherliquor supplemented with 20% (v/v) glycerol.

Data were collected under cryogenic conditions at P13, Deutsches Elektronen-Synchrotron. Data were processed using XDS and scaled with XSCALE[53]. All structures were determined by molecular replacement with PHASER[54], manually built in WinCOOT (v.0.9.6)[55] and refined with PHENIX (v.1.19.2)[56]. The search model for the structure was the hexameric Δ2–6 variant. Images of the structure were generated using PyMOL (v.2.5.2).

## Cryo-EM

For cryo-EM sample preparation, 4.5 µl of the protein sample (22.5 µM) was applied to glow-discharged Quantifoil 2/1 grids, blotted for 4 s with force 4 in a Vitrobot Mark III (Thermo Fisher) at 100% humidity and 4 °C, and plunge frozen in liquid ethane, cooled by liquid nitrogen. Cryo-EM data were acquired with a FEI Titan Krios transmission electron microscope (Thermo Fisher) using SerialEM software[57]. Movie frames were recorded at a nominal magnification of ×29,000 using a K3 direct electron detector (Gatan). The total electron dose of about 55 electrons per Å$^2$ was distributed over 30 frames at a pixel size of 1.09 Å. Micrographs were recorded in a defocus range from −0.5 to −3.0 µm.

## Image processing, classification and refinement

For the SeCS 18mer, all processing steps were carried out in cryoSPARC (v.3.2.0)[58]. A total of 1,408 movies were aligned using the patch motion correction tool, and contrast transfer function (CTF) parameters were determined using the patch CTF tool. An initial set of 10,173 particles were acquired through several rounds of blob picking, 2D classification and template picking for training a Topaz convolutional neural network particle picking model[59]. From all the corrected micrographs, 273,259 particles were extracted in a box size of 350 by 350 pixels at a pixel size of 1.09 Å using the Topaz extract tool together with the trained model. Overall, 224,041 particles were selected for the ab initio reconstruction after removing poor particles through 2D classification. The initial density map was then three-dimensionally (3D) classified and refined using the heterogenous refinement tool, which resulted in three classes. The dominant class (56.7% particles) was subjected to another round of heterogenous refinement, which led to two classes. A 3D non-uniform refinement of the main class (79.8% particles) imposing a C3 symmetry, followed by a local CTF refinement produced a final resolution of 3.93 Å (GSFSC = 0.143), which was used for model building. Local resolution of the density map was calculated with the local resolution estimation tool.

For the Δ2–6 sample, cryo-EM micrographs were processed on the fly using the Focus software package[60] if they passed the selection criteria (iciness < 1.05, drift 0.4 Å < $x$ < 70 Å, defocus 0.5 µm < $x$ < 5.5 µm, estimated CTF resolution < 6 Å). Micrograph frames were aligned using MotionCor2 (ref. 61) and the CTF for aligned frames was determined using GCTF[62]. From 5,419 acquired micrographs 1,687,951 particles were picked using the Phosaurus neural network architecture from crYOLO[63]. Particles were extracted with a pixel box size of 256 scaled down to 96 using RELION (v.3.1)[64] and underwent several rounds of reference-free 2D classification. Overall, 1,271,457 selected particles (Δ2–6) were re-extracted with a box size of 256 and imported into Cryosparc (v.2.3)[58]. For each sample, ab initio models were generated and passed through heterogeneous classification and refinement. Selected particles were re-imported to RELION and underwent several rounds of refinement, CTF-refinement (estimation of anisotropic magnification, fit of per-micrograph defocus and astigmatism and beamtilt estimation) and Bayesian polishing[65]. Final C1 refinement produced models with an estimated resolution of 3.1 Å for Δ2–6 (gold standard FSC analysis of two independent half-sets at the 0.143 cut-off). Local resolution and 3D FSC plots were calculated using RELION and the "Remote 3DFSC Processing Server" web interface[66], respectively.

For the H369R SeCS 54mer and 18mer, all processing steps were carried out in cryoSPARC (v.4.4.0). In total, 29,126 movies were aligned using the patch motion correction tool, and CTF parameters

were determined using the patch CTF tool. Next, 8,583 micrographs of estimated CTF fit ≤ 3.5 Å were selected for subsequent analysis. A Topaz particle picking model was generated by running several rounds of Topaz train and Topaz extract from an initial set of 150 manually picked particles. A total of 95,268 particles were picked using the trained model and extracted in a box size of 1,200 by 1,200 pixels at a pixel size of 0.79 Å. The particles were downsampled to a pixel size of 1.58 Å before 2D classification. 2D classes corresponded to the SeCS 54mer were selected to reconstruct two densities map using the ab initio reconstruction tool. All the extracted particles were re-aligned and 3D classified by running the heterogenous refinement tool using the density map corresponded to an intact 54mer as a reference. The 3D class (18.0% particles) was further refined by non-uniform refinement, which resulted in a final resolution of 5.91 Å (GSFSC = 0.143), which was used for model building. To reconstruct the mutant 18mer, 899,109 particles were picked using a 2D class corresponding to the 18mer as a template and extracted in a box size of 500 by 500 pixels at a pixel size of 0.79 Å. A total of 552,353 particles were selected from 2D classification to generate 3 initial maps. The major class was 3D classified and aligned, followed by a non-uniform refinement to produce the final 18mer density at 3.34 Å (GSFSC = 0.143). Local resolution of the density map was calculated with the local resolution estimation tool, and preferred orientation was assessed using the orientation diagnostics tool.

For 18meric SeCS, initial models were generated separately from their protein sequences using alphaFold[67] and then fitted as rigid bodies into the density using UCSF Chimera. The model was manually rebuilt using WinCoot (v.0.9.6)[55]. Non-crystallographic symmetry constraints were manually defined in PHENIX (v.1.19.2)[56] so that each monomer within one hexamer is linked to the two corresponding monomers in the other two hexamers (corresponding to a C3 symmetric refinement of the 18mer). For the Δ2–6 hexamer, a hexameric subunit was extracted from the 18mer model as a starting model for refinement. The model was firstly rigid-body fitted into the density, and manually refined in WinCoot (v.0.9.6)[55]. Both models were subjected to real-space refinements against the respective density maps using phenix.real_space_refine implemented in PHENIX (v.1.19.2)[56]. Images of the structures were generated using PyMOL (v.2.5.2). For the 54mer structure of SeCS H369R, we used the dimers extracted from the WT 18mer structure as our starting model and fitted them as rigid bodies into the density using UCSF Chimera. We then truncated all side chains using pdbtools within PHENIX (v.1.19.2)[56]. The structure was then subjected to one round of real space refinement using default parameters in PHENIX. For the 18mer structure of SeCS H369R, we also used the 18mer SeCS structure as the starting model. Individual dimers were first fitted as rigid bodies into the density using UCSF Chimera. We then subjected the structure to one round of flexible fitting with default parameters, followed by refinement with default parameters using the Namdinator server[68]. The model was then manually rebuilt in WinCoot (v.0.9.6)[55]. In this model, we truncated the substrate lids (residues 220–312, which are not part of the fractal interface) in all chains owing to poorly resolved density in our map, which made it difficult not to introduce register errors during refinement.

## SAXS data collection and analysis

SAXS experiments were carried out at the BM29 beamline at the ESRF[69] using a PILATUS3X 2M photon counting detector (DECTRIS) at a fixed distance of 2,827 m. Protein samples were prepared in 25 mM Tris-HCl buffer pH 7.5 and 200 mM NaCl as a dilution series. Buffer matching was achieved by dialysis and all measurements were carried out at 20 °C. The sample delivery and measurements were performed using a 1 mm diameter quartz capillary, which is part of the BioSAXS automated sample changer unit (Arinax). Before and after each sample measurement, the corresponding buffer was measured and averaged. A total of ten frames (one frame per second) were taken for each sample. All experiments were conducted with the following parameters: beam current of 200 mA; flux of $2.6 \times 1,012$ photons s$^{-1}$ at sample position; wavelength of 1 Å; and estimated beam size of $200 \times 200$ μm. Processing and analysis of collected SAXS data were performed using ScÅtter IV[70]. The $R_g$ was determined by Guinier approximation. Plotting of the SAXS profiles and Guinier regions used BioXTAS RAW[71].

## Construction of atomic models of the 54mers using 18mers

We used the align, translate, and rotate commands within PyMOL (v.2.5.2) to model how a 54mer complex would assemble if the 4.0° and 4.2° dimer rotations and 60° dihedral angle between dimers that are observed in the 18mer structure were applied. The rotation was applied to the connecting dimers of the three 18mer-subcomplexes that built the 54mer. To do this, copies of the hexamers that constitute the 18mer were rotated by 120°, so as to overlay the corner dimers by edge dimers. Two 18mer copies were subsequently connected to the rotated corners by two steps of structural alignment, which placed the residues that should form the third interface 210 Å from each other (Extended Data Fig. 2g).

## Calculation of $R_g$ values

Calculation of $R_g$ values was done using gmx gyrate from the GROMACS 2022.2 simulation package[72] from the atomic models of the 6mer, 18mer and the 54mer.

## Displacement vectors, rotational axes and dihedral angles of atomic models

Symmetry axes were generated with AnAnaS[73], and rotation axes and angles were calculated using PyMOL (v.2.5.2) and a compatible script. Displacement vectors were drawn between Cα atoms of the aligned structures using the object argument and cgo-arrow. Dihedral angles between dimers across the fractal interface were calculated in PyMOL (v.2.5.2). The centre of mass of both dimers, as well as of one monomer from each dimer, was first calculated with the com command. The dihedral angle was then calculated using get_dihedral along the axis defined by the two centres of mass of the dimers.

## Reporting summary

Further information on research design is available in the Nature Portfolio Reporting Summary linked to this article.

## Data availability

Atomic structures reported in this paper have been deposited into the Protein Data Bank under accession codes 8AN1, 8BP7, 8BEI, 8RJK and 8RJL. The cryo-EM data have been deposited into the Electron Microscopy Data Bank under accession identifiers EMDB-15529, EMDB-16004, EMDB-19250 and EMDB-19251. All raw data for MP spectra, growth curves and kinetic traces as well as phylogenetic trees, alignments and ancestral sequences have been deposited into Edmond, the Open Research Data Repository of the Max Planck Society, for public access and available under https://doi.org/10.17617/3.KNEQIR (ref. 74). NCBI reference sequence accession codes for the protein sequences that were experimentally investigated are provided in the Supplementary Information (Supplementary Table 3). All NCBI reference sequence accession codes for protein sequences that were used for the evolutionary analysis are available from the multiple sequence alignment that is deposited in the Edmond repository.

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

**Acknowledgements** F.L.S., G.K.A.H., T.J.E., D.S. and N.P. were supported by the Max Planck Society. S.B. was supported by the Helmholtz Association. L.J.P. and E.G.M. were supported by a project grant from the Swedish Research Council (2020-04825). J.L.P.B. acknowledges support from the Leverhulme Trust (grant no. RPG-2021-246). For the purpose of Open Access, the author has applied a CC BY public copyright licence to any Author Accepted Manuscript (AAM) version arising from this submission. K.F. was supported by DFG grant Fo195/16-2. This work was supported by the Max Planck Society within the framework of the MaxGENESYS project (D.S.) and the International Max Planck Research School for Principles of Microbial Life: from molecules to cells, from cells to interactions (M.d.C.S.O.). J.M.S. acknowledges the DFG for an Emmy Noether grant (SCHU 3364/1-1). This research was co-funded by the European Union (G.K.A.H. and F.L.S.: ERC, EVOCATION, 101040472; J.M.S.: ERC, Two-CO2-One, 101075992; G.B. and C.-N.M.: ERC, KIWIsome, 101019765). The views and opinions expressed are, however, those of the author(s) only and do not necessarily reflect those of the European Union or the European Research Council. Neither the European Union nor the granting authority can be held responsible for them. The authors acknowledge and thank staff at EMBL Hamburg at the PETRA III storage ring (DESY) and staff at the European Synchrotron Radiation Facility (ESRF) for support; M. Tully and A. Popov at BM29 for their support with SAXS data acquisition; M. Schauflinger from the EM Facility of the Institute of Virology, Philipps University Marburg for access to their electron microscope during times when we had technical issues with our system; staff at the cryo-EM Facility of the Philipps University of Marburg for their contribution; A. Kumar and M. Girbig for discussions and assistance with data processing; and P. Pfister, R. Inckemann, S. Murray and M. Gottfried for discussions and critical reading of the manuscript.

**Author contributions** F.L.S. and G.K.A.H. conceived the project, analysed data and planned experiments. J.M.S. contributed to project conceptualization and planned and supervised the EM work. F.L.S. performed molecular work, phylogenetics, ancestral sequence reconstruction, protein purification, MP measurements, SAXS measurements, enzyme kinetic measurements, in vivo experiments of *S. elongatus* and analysis of the structural data. Y.K.L. and F.L.S processed the cryo-EM datasets. Y.K.L. refined the cryo-EM structures. T.H. performed and analysed negative-stain EM data. S.B. collected and processed cryo-EM datasets. L.J.P. performed in silico structural analysis and measurements with supervision from E.G.M. C.-N.M. collected, solved and refined the X-ray structure with supervision from G.B. W.S. performed native mass spectrometry with supervision from J.L.P.B. N.P. performed and analysed liquid chromatography–tandem mass spectrometry measurements. M.d.C.S.O. and D.S. supported the construction of *S. elongatus* strains. E.N., K.F. and T.J.E. supported in vivo work with *S. elongatus* practically and intellectually. F.L.S. and G.K.A.H. wrote the manuscript with contributions and comments from all authors.

**Funding** Open access funding provided by Max Planck Society.

**Competing interests** The authors declare no competing interests.

**Additional information**
**Correspondence and requests for materials** should be addressed to Jan M. Schuller or Georg K. A. Hochberg.

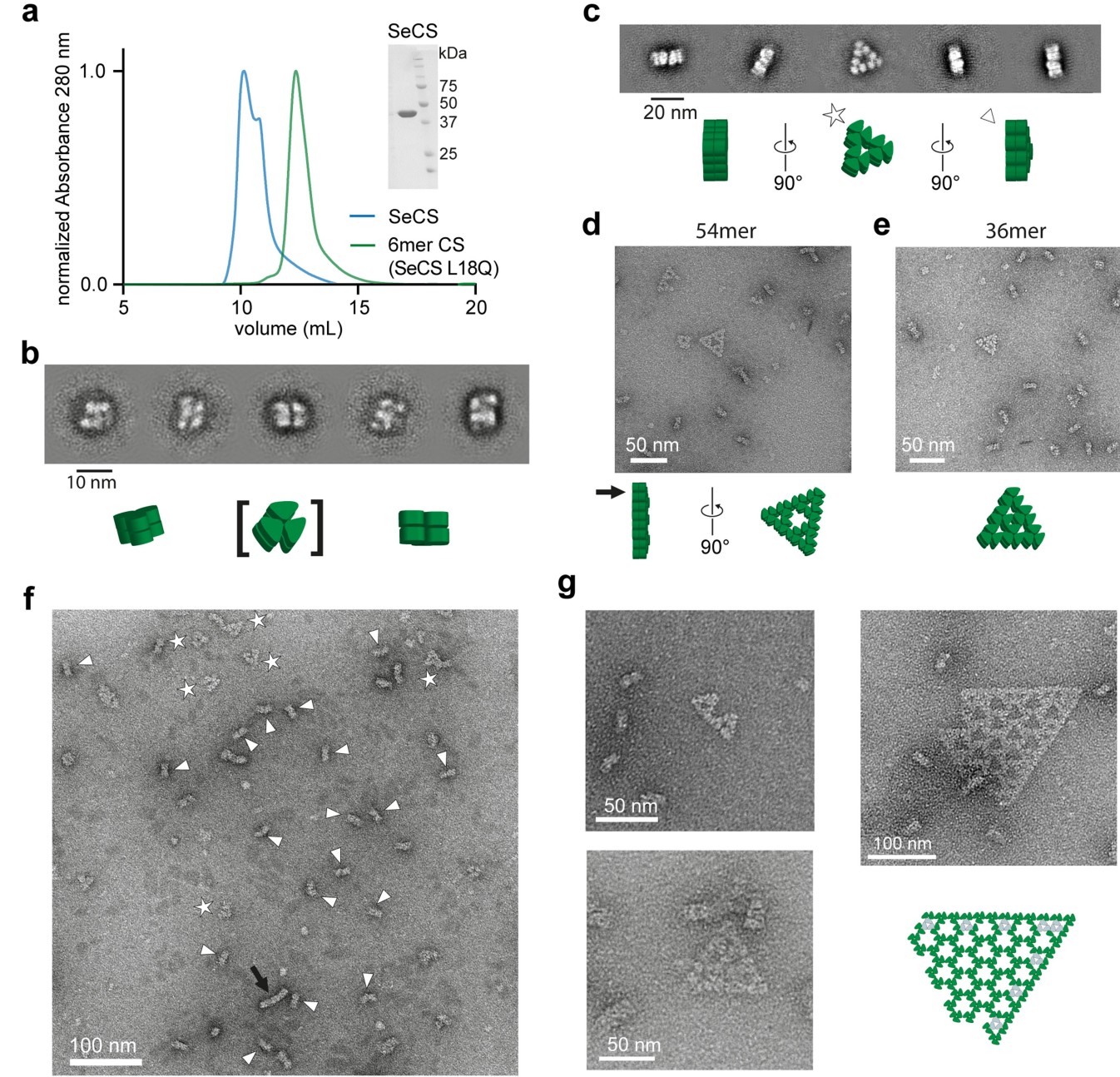

**Extended Data Fig. 1 | SeCS forms complexes of 18 subunits and larger assemblies. (a)** Size exclusion chromatography profile of purified WT SeCS and a 6mer SeCS-variant (SeCS L18Q, see Fig. 5d). The size exclusion chromatography runs were performed three independent times for both samples with similar results. Inlay shows a SDS-PAGE gel of the purified WT SeCS (Uncropped image in Supplementary Fig. 1a). **(b)** 2D class averages from negative stain electron microscopy of a 6mer SeCS variant (SeCS L18Q, s. Fig. 5d) which yielded different particle orientations but no top views. **(c)** 2D class averages from negative stain electron microscopy of the 18mer SeCS complexes. Strong preferential orientation towards side views, where the complex lays on the edge or tip of the triangle. Compare also cryo-EM 2D class averages supplementary Fig. 3. **(d-f)** Detail from example micrographs from

negative stain electron microscopy. For the depicted structures we observed for the 18mer = 1491 particles, 54mer = 200 particles and the 36mer = 186 particles. Symbols (star, triangle, arrow) indicate the respective complex and orientation in the overview micrograph (f). **(g)** Additional assemblies that were observed only 2–4 times from a total of 20 micrographs (4096×4096 pixels). The largest shown assembly was observed only a single time but we created a model based on hexameric subcomplexes, see below the micrograph. Subunits colored in grey rely on a three-way junction of dimers and subunits colored in green rely on a two-way junction. For 18mers and 54mers all connections are two-way junctions. The central connection of 36mers is in contrast a three-way junction.

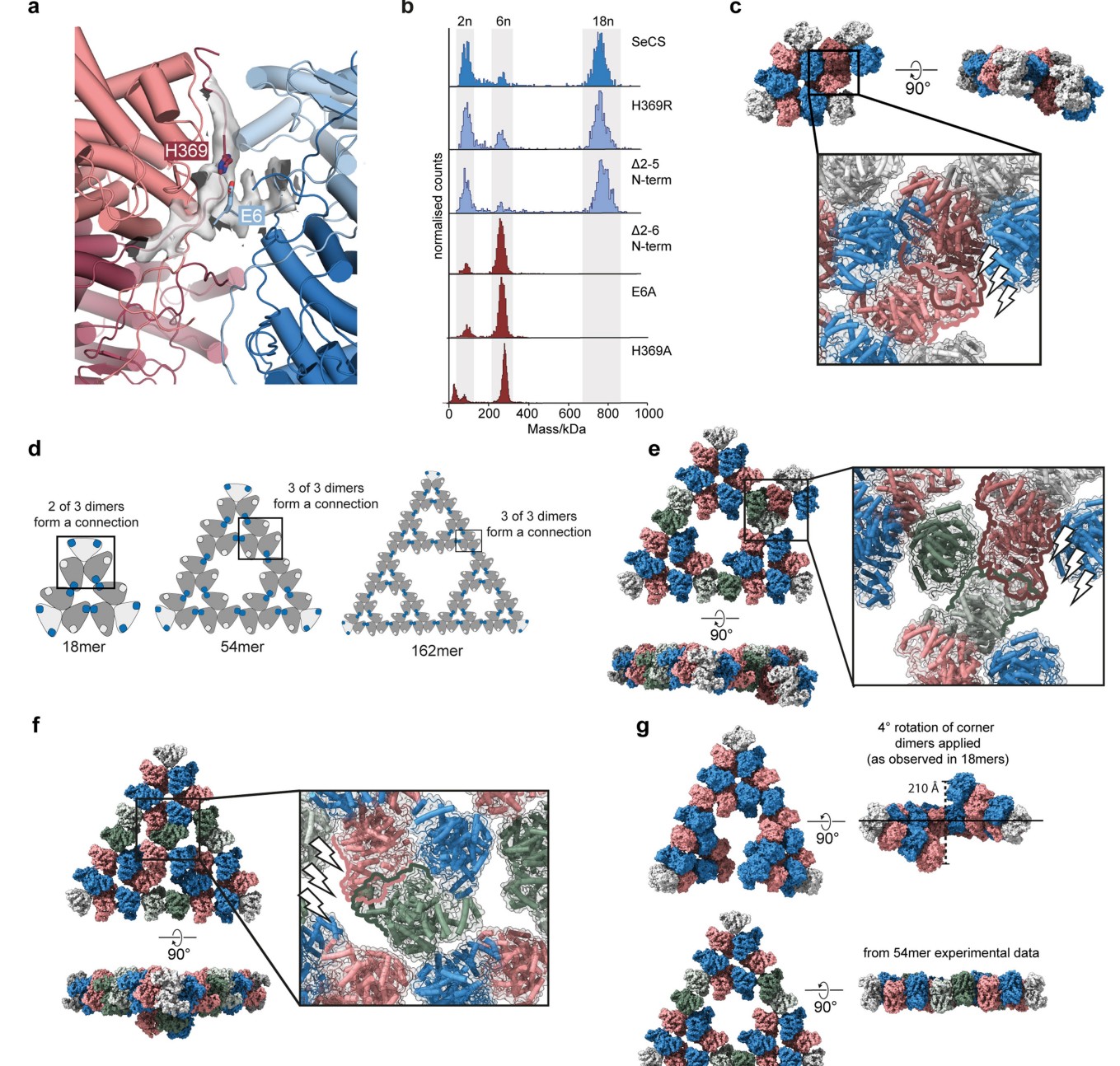

**Extended Data Fig. 2 | Interface residues of the 18mer and construction of 54mers.** (**a**) Close-up of the 18mer cryo-EM density at the interface that connects hexamers with key residues annotated. (**b**) MP measurements of variants of SeCS. (**c**) Addition of a hexamer to the edge of an 18mer via the interface residues that do not participate in the fractal connection (dark-red dimer from hexamer binding to a blue dimer from 18mer). The angle of this interaction would force the added hexamer out of the plane of the 18mer. The interaction would also introduce steric clashes with the salmon-colored dimer of the 18mer. (**d**) Schematic depiction of different interactions of the hexameric subcomplexes within subsequent levels of Sierpińksi triangles. Addition of a hexamer to the edge (**e**) or central void (**f**) of an 54mer connecting to the interface residues that do not participate in the fractal connection via the same interaction that is observed between 18mers. In both cases the added hexamer tilts out of the plane of the 54mer and introduces steric clashes with a dimer within the 54mer. (**g**) Formation of 54mer from the 18mer structures. The 18mer is a flat, closed triangle because of the internal rotation introduced by the interface of the connecting subunits. If the corner dimers of the 18mers are rotated by the same 4° rotation when forming a 54mer the angle between 18mer subcomplexes is too large and does not form planar, closed triangles. Our empirical density of the closed 54mer is shown in comparison.

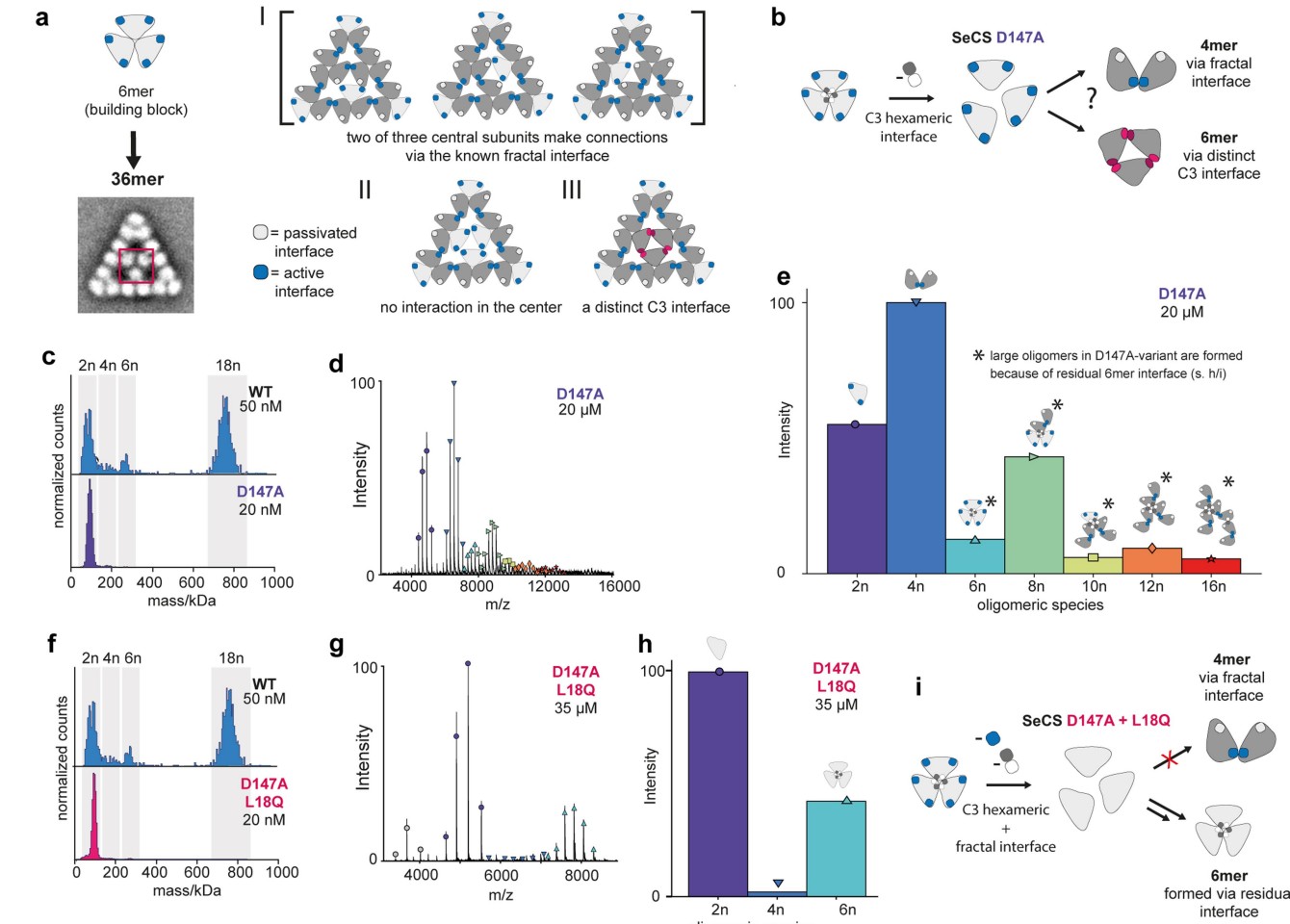

**Extended Data Fig. 3 | 36mer complexes are not stabilized via an additional C3 interface. (a)** Pascal's triangle-like 36mer complexes contain a three-fold connection in their center that is not observed in fractal-like 18mers and 54mers. From the 18mer structure a three-way interaction via the observed interface is not possible as it passivates the subunits. Therefore, either only two subunits can connect in the center (I), no subunits interact (II) or there is a distinct C3-interface that allows for a threefold interaction (III). **(b)** The mutation D147A destabilizes the interface connecting three dimers to a hexamer, that forms the building block of all larger oligomers. The interface that induces fractal assembly is unchanged and allows for formation of 4mers. In case of an additional distinct C3-interface, the formation of stable 6mers is expected. **(c)** Mass photometry measurements of the D147A variant reveal dimers and the disruption of the hexamer interface. **(d)** Native mass spectrometry of D147A SeCS at high protein concentration (20 µM). **(e)** The distribution of oligomers determined from (d) revealed a strong preference for the formation of 4mers. A low

abundance of 6mers renders an additional distinct C3-interface as unlikely or at least much less stable. Larger oligomers arise probably due to an incomplete disruption of the hexamer subcomplex interface. Cartoons indicate potential structures that correspond to the larger observed oligomers. **(f)** MP measurement of an additional variant in which the fractal interface was also disrupted (SeCS D147A + L18Q). This variant showed to form only dimers at nanomolar concentrations using MP. **(g)** Native mass spectrometry of D147A + L18Q SeCS at high protein concentration (35 µM). **(h)** The distribution of oligomers determined from (g) revealed the formation of mostly dimers and hexamers. **(i)** The formation of hexamers in the variant D147A + L18Q SeCS additionally supports that the hexamer interface was not completely destroyed by the mutant D147A and that the larger oligomers (≥ 8mer, d + e) are formed because of residual affinity in the hexamer interface, not because of an additional C3 interface.

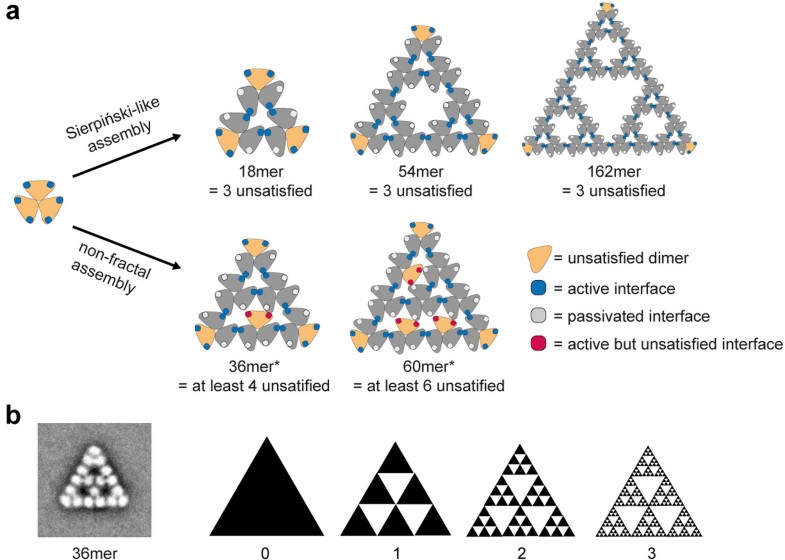

**Extended Data Fig. 4 | Interface occupancy in fractal and non-fractal assemblies. (a)** Assembly into Sierpiński-triangle complexes from hexameric subcomplexes always results in only three unsatisfied dimers at the corners of the triangle. When assembled into other complexes e.g. the 36mers or larger forms of lattice-like triangles at least 4 or more dimers stay unsatisfied.

\* The active but unsatisfied interfaces in the inner part of the represented structures can be located at different positions, similar to a resonance structure. Depicted is one possibility. **(b)** Schematic depiction of a fractal pattern that can be assembled from 36mers.

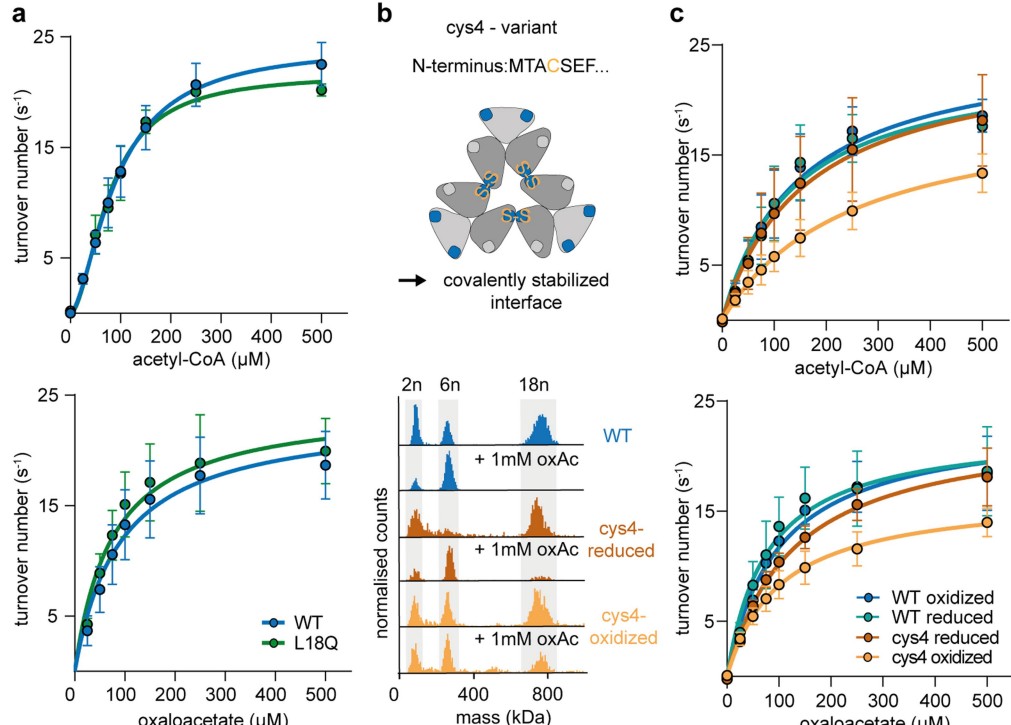

**Extended Data Fig. 5 | Enzyme kinetics of SeCS and its variants. (a)** Michaelis-Menten kinetics of SeCS and the hexameric L18Q variant. Data presented as mean values, error bars = SD, n = 3 biological replicates with 3 technical replicates each. **(b)** Schematic depiction of the cys4-variant of SeCS, which stabilizes 18mer-complexes by a reversible disulfide bridge in the fractal interface and prevents the disassembly at high substrate concentration. MP measurements of SeCS and the cys4-variant under oxidizing and reducing conditions, with and without oxaloacetate (oxAc). **(c)** Michaelis-Menten kinetics of SeCS and cys4-variant after oxidation and subsequent reduction. Data presented as mean values, error bars = SD, n = 3 biological replicates with 3 technical replicates each.

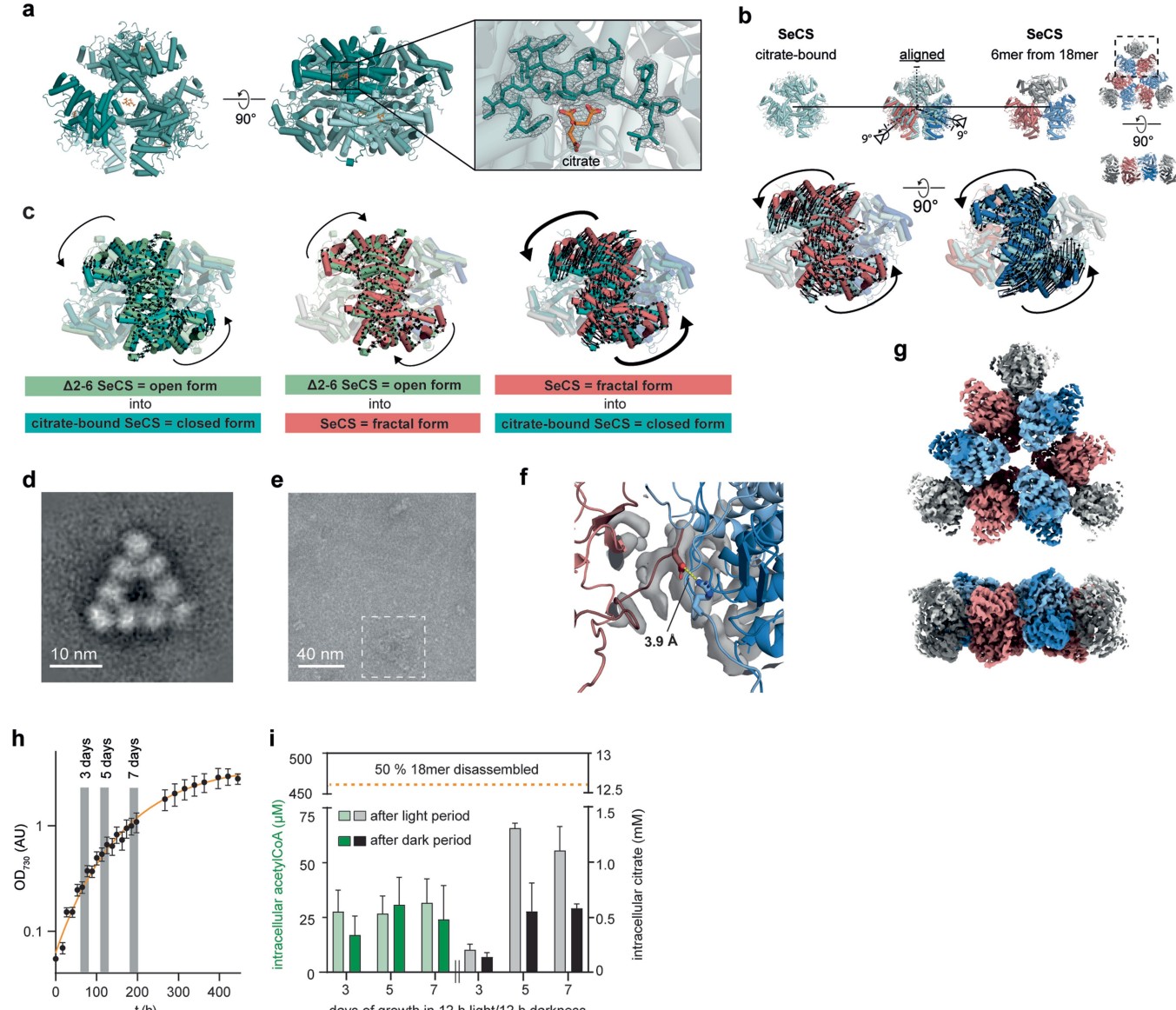

**Extended Data Fig. 6 | Structural changes in the citrate-bound SeCS, structural integrity of the H369R variant, and intracellular CS metabolite concentrations.** (**a**) Molecular model of hexameric SeCS bound to citrate, solved by X-ray crystallography to a resolution of 2.7 Å. Zoom displays the density of a citrate-molecule inside the substrate binding pocket of the enzyme. (**b**) Alignment of the citrate-bound structure with a hexameric subcomplex from the 18mer structure. Arrows depict the molecular displacement from the 18mer to the citrate-bound structure, which can be described by a 9° rotation around an internal axis within the dimer-subcomplexes (black axis). The symmetry axes of the dimer-interfaces are shown as dotted lines for reference. (**c**) Comparison of the conformational changes between Δ2-6 SeCS (representing the typical open form of CS structures[24]), citrate-bound SeCS (representing the closed form of CS) and the fractal form of 18meric SeCS. (**d**) Negative stain 2D class average of the 18mer formed by H369R SeCS at 450 nM. (**e**) Detail of a negative stain micrograph showing a 54mer formed by H369R SeCS at 450 nM.

We collected 196 micrographs (2048×2048 pixel) in total. (**f**) Close up on the interaction between R369 and E6 in the 18mer structure H369R SeCS. (**g**) Cryo-EM density of an 18mer from the H369R SeCS variant resolved to 3.5 Å. (**h**) Growth curve of *S. elongatus* PCC 7942 cultivated under circadian cycles (12 h light and 12 h darkness). Grey columns indicate the growth phases, in which samples were taken for metabolomic analysis. Cultures were set up in three biological replicates, data are presented as mean values and error bars indicate SD. (**i**) Intracellular concentration of metabolites in *S. elongatus* grown under circadian conditions. Samples were taken at the end of a full dark or light cycle, respectively. Oxaloacetate concentrations could not be measured due to low abundance and stability but are thought to be extremely low[75]. The concentration values that induce the disassembly of 18mers are taken from the titrations in Fig. 4a. Data are presented as mean values, error bars indicate SD from three biological replicates.

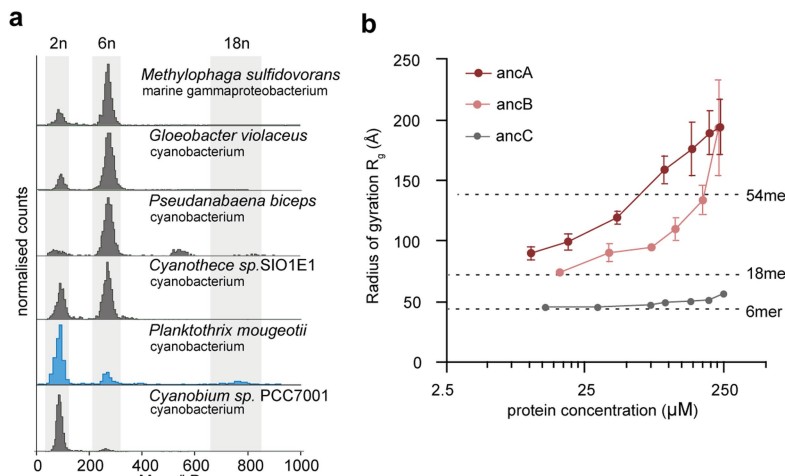

**Extended Data Fig. 7 | Oligomeric state of extant cyanobacterial CSs and ancestral CSs.** (**a**) MP measurements of purified CS from extant cyanobacteria. Assembly into fractal 18mers was only detected in the CS of *P. mougeotii*. The CS from *Cyanobium* sp. PCC7001, which belongs to the immediate sister-group of *S. elongatus* formed only dimers. The fractal assembly was therefore lost in this lineage. (**b**) $R_g$ measurements for ancA-C at varying protein concentrations based on SAXS measurements. One sample for each concentration step was measured over 10 frames. The data presented is the inferred $R_g$ value using Guinier approximation and error bars correspond to the standard deviation of fit values calculated from the covariance matrix (ScÅtter IV). Dashed lines indicate $R_g$s calculated from structural models of the 6mer, 18mer, and 54mer.

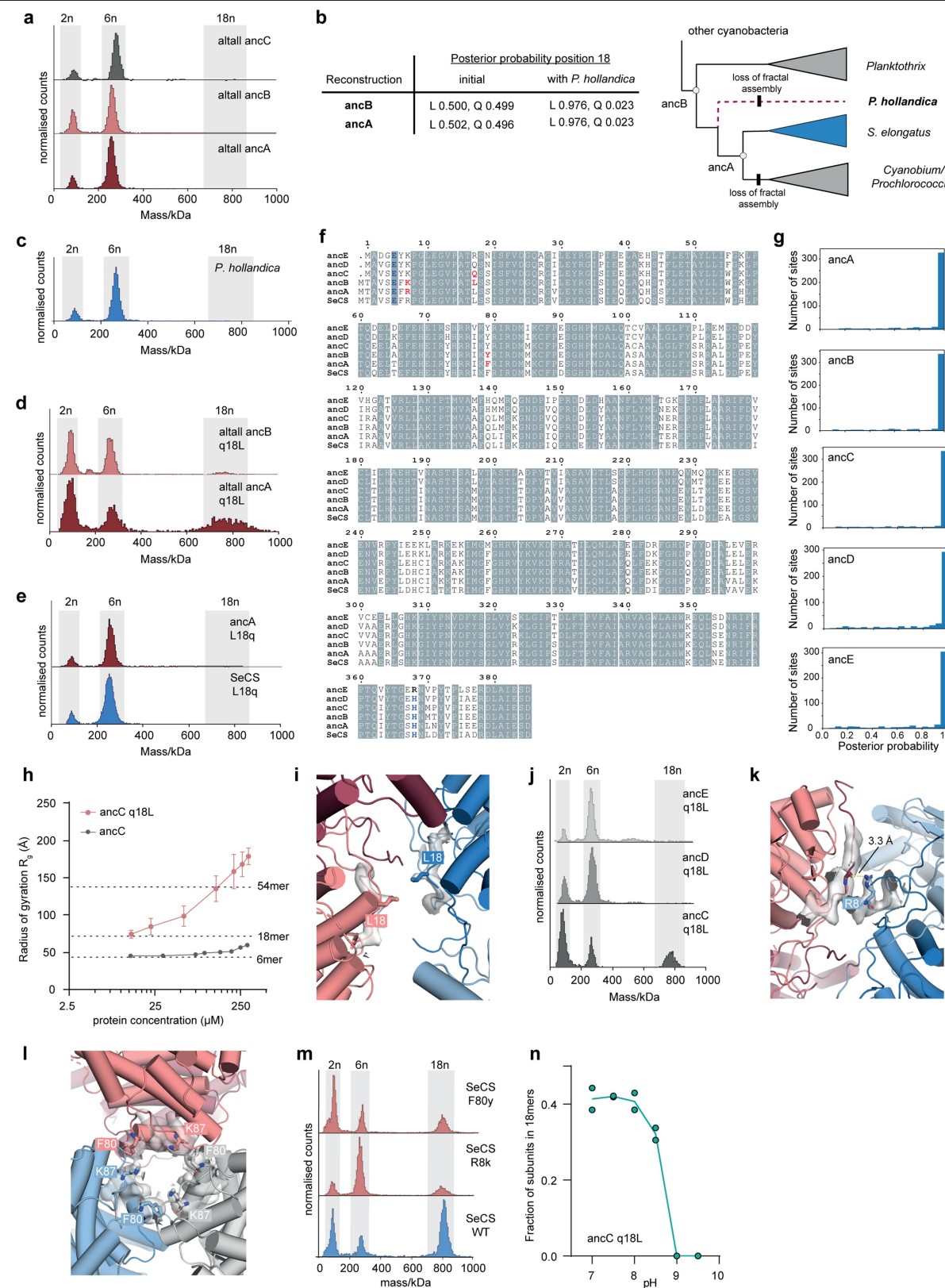

**Extended Data Fig. 8** | See next page for caption.

**Extended Data Fig. 8 | Alternative reconstructions of the ancestral proteins and emergence of fractal assembly.** (**a**) MP measurements of purified alternative ancestral proteins (altall = all position changed to the second most likely amino acid, if PP > 0.2). (**b**) Posterior probability for the position 18 of ancA-B. The initial reconstruction was very ambiguous about this state reconstructing Q or L with similar probabilities. The alternative ancestors therefore contained a Q at position 18. The sequence from *P. hollandica* was added as sister to *S. elongatus* and the *Cyanobium-/Prochloroccous* group to the phylogeny, which is well established in cyanobacterial species trees (see methods). Subsequent ancestral sequence reconstruction with the modified alignment shifted the probability strongly towards leucine at position 18. Indicated on the tree are the losses of fractal assembly on two branches, which were observed from the assembly state of extant CS from *P. hollandica* and *Cyanobium* (**c**) MP measurement of the purified CS from *P. hollandica*. (**d**) We adjusted the altall sequences of ancA-B to include q18L corresponding to the reconstruction including *P. hollandica*. MP measurements of purified modified altall ancA-ancB showed assembly into 18mers supported the inference of ancestral assembly states of ancA-B. (**e**) MP measurements of ancA and SeCS with a reversal to L18q, which prevents assembly into 18mers. (**f**) Alignment of the inferred amino acid sequences of the ancestral proteins ancA-E and SeCS.

Interface residues E6 and H369 are colored in blue and historical changes that were found to have had an influence on the assembly into fractals are colored in red (residues 8, 18, 80). (**g**) Histograms that display the distribution of the posterior probabilities of the maximum a posterior state across reconstructed sites for all five ancestral proteins. (**h**) $R_g$ measurements ancC q18L at varying protein concentrations based on SAXS measurements. One sample for each concentration step was measured over 10 frames. The data presented is the inferred $R_g$ value using Guinier approximation and error bars correspond to the standard deviation of fit values calculated from the covariance matrix (ScÅtter IV). Dashed lines indicate $R_g$s calculated from structural models of the 6mer, 18mer, and 54mer. (**i**) Close-up of cryo-EM density of SeCS with key substitution L18 annotated. (**j**) MP measurements of ancC-E variants with the q18L substitution. The substitution only triggers the formation of 18mers when introduced into ancC. Close-up of cryo-EM density of SeCS with key substitutions R8 (**k**) and F80 (**l**) annotated. (**m**) MP measurements of SeCS variants in which the identified important historical substitutions between ancB and ancA were reversed (F80y, R8k). (**n**) MP quantification of the fraction of CS monomers in 18mers at different pH values for ancC q18L. Two independent measurements were performed for each pH value.

**Extended Data Table 1 | Cryo-EM data collection, refinement, and validation statistics**

| | SeCS 18mer (PDB 8AN1) (EMDB-15529) | Δ2-6 6mer (PDB 8BEI) (EMDB-16004) | H369R 18mer (PDB 8RJL) (EMDB-19251) | H369R 54mer (PDB 8RJK) (EMDB-19250) |
|---|---|---|---|---|
| **Data collection and Processing** | | | | |
| Microscope | Titan Krios | Titan Krios | Krios G4 | Krios G4 |
| Voltage (keV) | 300 | 300 | 300 | 300 |
| Camera | Gatan K3 | Gatan K3 | Falcon 4 | Falcon 4 |
| Magnification | 29,000X | 29,000X | 29,000X | 29,000X |
| Pixel size at detector (Å/pixel) | 1.09 | 1.09 | 0.79 | 0.79 |
| Total electron exposure (e$^-$/Å$^2$) | 55 | 55 | 60 | 60 |
| Exposure rate (e-/pixel/sec) | 4.5 | 4.5 | 4.5 | 4.5 |
| EER fractionation (frames) | | | 50 | 50 |
| Number of frames collected during exposure | 30 | 30 | | |
| Defocus range (µm) | -3.0 to -0.5 | -3.0 to -0.5 | -3.0 to -0.5 | -3.0 to -0.5 |
| Automation software | SerialEM | SerialEM | SerialEM | SerialEM |
| Tilt angle (°) | 20 | N/A | 20 | 20 |
| Micrographs collected (no.) | 1,408 | 5,419 | 29,126 | 29,126 |
| Micrographs used (no.) | 1,408 | 5,419 | 8,583 | 8,583 |
| Total extracted particles (no.) | 273,259 | 1,687,951 | 899,109 | 95,268 |
| **Reconstruction** | | | | |
| Refined particles (no.) | 224,041 | 1,271,457 | 552,353 | 95,268 |
| Final particles (no.) | 102,024 | 774,026 | 178,751 | 17,191 |
| Symmetry imposed | C3 | C3 | C3 | D3 |
| Resolution (global, Å) | | | | |
| FSC 0.5 (unmasked/masked) | 6.29/4.26 | 3.53/3.32 | 4.3/3.67 | 19.5/8.5 |
| FSC 0.143(unmasked/masked) | 4.30/3.93 | 3.28/3.06 | 3.8/3.3 | 9.4/5.9 |
| Resolution range (local, Å) | 3.4 – 8.7 | 2.92 – 4.41 | 10.0-1.68 | 69.3-3.4 |
| Map sharpening *B* factor (Å$^2$) | -174.4 | -139.2 | -118.8 | -218.5 |
| Map sharpening methods | Global B factor | Global B factor | Global B factor | Global B factor |
| **Model composition** | | | | |
| Protein | SeCS | Δ2-6 | H369R | H369R |
| **Model Refinement** | | | | |
| Real space refinement software | PHENIX v1.19.2 | PHENIX v1.19.2 | PHENIX v1.20.1 | PHENIX v1.20.1 |
| Model-Map scores (CC mask) | 0.81 | 0.83 | 0.76 | 0.66 |
| *B* factors (Å$^2$) (min/max/mean) | 30.00/319.81/105.08 | 0.08/62.31/26.48 | 8.93/193.69/60.68 | 30.00/997.6/572.25 |
| Protein residues | 6255 | 2100 | 4575 | 19548 |
| R.m.s. deviations from ideal values | | | | |
| Bond lengths (Å) | 0.004 | 0.003 | 0.012 | 0.004 |
| Bond angles (°) | 0.919 | 0.544 | 1.492 | 1.032 |
| **Validation** | | | | |
| MolProbity score | 1.94 | 1.58 | 2.70 | 1.65 |
| CaBLAM outliers | 2.18 | 2.4 | 4.84 | 3.39 |
| Clashscore | 6.16 | 5.76 | 11.08 | 3.98 |
| Poor rotamers (%) | 3.10 | 1.38 | 4.59 | 0.00 |
| Cβ outliers (%) | 0.00 | 0.00 | 0.28 | 0.00 |
| EMRinger score | 1.35 | 2.57 | 1.32 | N/A |
| **Ramachandran plot** | | | | |
| Favored (%) | 96.5 | 97.08 | 86.37 | 92.67 |
| Allowed (%) | 3.40 | 2.92 | 12.55 | 7.19 |
| Outliers (%) | 0.10 | 0.00 | 1.08 | 0.14 |

**Extended Data Table 2 | Data collection and refinement statistics for the crystal structure of citrate bound SeCS**

| | SeCS - citrate bound (PDB 8BP7) |
|---|---|
| Space group | $P6_122$ |
| **Cell dimensions** | |
| $a, b, c$ (Å) | 170.46 170.46 545.44 90 90 120 |
| a,b,g (°) | 90 90 120 |
| Wavelength (Å) | 0.976260 |
| Resolution (Å) | 49.21 - 2.71 (2.807 - 2.71) |
| $R_{merge}$ | 0.4252 (5.392) |
| $R_{pim}$ | 0.0765 (1.077) |
| $R_{meas}$ | 0.4323 (5.504) |
| $I / \sigma I$ | 9.19 (0.68) |
| Completeness (%) | 99.90 (99.98) |
| Redundancy | 28.6 (22.9) |
| $CC1/2$ | 0.998 (0.25) |
| **Refinement** | |
| Resolution (Å) | 49.21 - 2.71 (2.74 - 2.71) |
| No. reflections | 127399 (12503) |
| $R_{work} / R_{free}$ | 0.25/0.28 |
| No. atoms | 17892 |
| Protein | 17571 |
| Ligand/ion | 172 |
| Water | 149 |
| $B$-factors (Å$^2$) | 75.27 |
| Protein (Å$^2$) | 75.21 |
| Ligand/ion (Å$^2$) | 93.27 |
| Water (Å$^2$) | 60.88 |
| **R.m.s. deviations** | |
| Bond lengths (Å) | 0.012 |
| Bond angles (°) | 1.76 |
| **Ramachandran** | |
| Favored (%) | 96.57 |
| Allowed (%) | 3.2 |
| Outliers (%) | 0.23 |

# Reporting Summary

## Statistics

For all statistical analyses, confirm that the following items are present in the figure legend, table legend, main text, or Methods section.

| n/a | Confirmed | |
|---|---|---|
| ☐ | ☒ | The exact sample size (*n*) for each experimental group/condition, given as a discrete number and unit of measurement |
| ☐ | ☒ | A statement on whether measurements were taken from distinct samples or whether the same sample was measured repeatedly |
| ☒ | ☐ | The statistical test(s) used AND whether they are one- or two-sided<br>*Only common tests should be described solely by name; describe more complex techniques in the Methods section.* |
| ☒ | ☐ | A description of all covariates tested |
| ☒ | ☐ | A description of any assumptions or corrections, such as tests of normality and adjustment for multiple comparisons |
| ☐ | ☒ | A full description of the statistical parameters including central tendency (e.g. means) or other basic estimates (e.g. regression coefficient) AND variation (e.g. standard deviation) or associated estimates of uncertainty (e.g. confidence intervals) |
| ☒ | ☐ | For null hypothesis testing, the test statistic (e.g. *F*, *t*, *r*) with confidence intervals, effect sizes, degrees of freedom and *P* value noted<br>*Give P values as exact values whenever suitable.* |
| ☒ | ☐ | For Bayesian analysis, information on the choice of priors and Markov chain Monte Carlo settings |
| ☒ | ☐ | For hierarchical and complex designs, identification of the appropriate level for tests and full reporting of outcomes |
| ☒ | ☐ | Estimates of effect sizes (e.g. Cohen's *d*, Pearson's *r*), indicating how they were calculated |

*Our web collection on statistics for biologists contains articles on many of the points above.*

## Software and code

Policy information about availability of computer code

| | |
|---|---|
| Data collection | Data collection was performed using the software provided by the respective instrument. Serial EM v3.8, AcquireMP v1.2.1, Tecan i-control 3.9.1. Amino acid sequences were collected from deposited sequences at the National Center for Biotechnology Information |
| Data analysis | ata analysis was performed using publicly available software as detailed in citations included in the manuscript and SI. MUSCLE v3.8.31 28, raxML v8.2.10, BOOSTER v0.1.0, PhyML 3.0, PAML v4.9, PAUP 4.0a, DiscoverMP v2.5.0, UniDec 4.0.2., GraphPad Prism 8.4.3, Excel Version 1808, Adobe Illustrator v24.0.2, MassHunter QQQ Quantative Analysis V10.0, cisTEM 1.0.0., XDS Version January 10, 2022; XSCALE Version January 10, 2022, PHASER 1.18.2-3874-000, WinCoot 0.9.6, PHENIX v1.19.2, cryoSPARC v3.2.0, v4.4.0 and v2.3, Gctf 1.06, Focus v1.0.0, RELION 3.1, ScÅtter IV, BioXTAS RAW 2.1.4, PyMOL 2.5.2, GROMACS 2022.2, AnAnaS v.0.6, Topaz v0.2.5, crYOLO v1.9.3, AlphaFold v2.1.2, UCSF Chimera v1.16, Namdinator v1.0 |

For manuscripts utilizing custom algorithms or software that are central to the research but not yet described in published literature, software must be made available to editors and reviewers. We strongly encourage code deposition in a community repository (e.g. GitHub). See the Nature Portfolio guidelines for submitting code & software for further information.

## Data

Policy information about availability of data

All manuscripts must include a data availability statement. This statement should provide the following information, where applicable:

- Accession codes, unique identifiers, or web links for publicly available datasets
- A description of any restrictions on data availability
- For clinical datasets or third party data, please ensure that the statement adheres to our policy

Atomic structures reported in this paper are deposited to the Protein Data Bank under accession codes 8AN1, 8BP7, 8BEI, 8RJK and 8RJL. The cryo-EM data were deposited to the Electron Microscopy Data Bank under EMDB-15529, EMDB-16004, EMDB-19250 and EMDB-19251. All raw data for MP spectra, growth curves and kinetic traces as well as phylogenetic trees, alignments, and ancestral sequences are deposited on Edmond, the Open Research Data Repository of the Max Planck Society for public access and available under https://doi.org/10.17617/3.KNEQIR. NCBI reference sequence accession codes for the protein sequences that were experimentally investigated are found in the Supplementary Information (Supplementary Table 3). All NCBI reference sequence accession codes for protein sequences that were used for the evolutionary analysis are found in the multiple sequence alignment that is deposited in the Edmond repository.

## Research involving human participants, their data, or biological material

Policy information about studies with human participants or human data. See also policy information about sex, gender (identity/presentation), and sexual orientation and race, ethnicity and racism.

| | |
|---|---|
| Reporting on sex and gender | NA |
| Reporting on race, ethnicity, or other socially relevant groupings | NA |
| Population characteristics | NA |
| Recruitment | NA |
| Ethics oversight | NA |

Note that full information on the approval of the study protocol must also be provided in the manuscript.

# Field-specific reporting

Please select the one below that is the best fit for your research. If you are not sure, read the appropriate sections before making your selection.

☒ Life sciences ☐ Behavioural & social sciences ☐ Ecological, evolutionary & environmental sciences

For a reference copy of the document with all sections, see nature.com/documents/nr-reporting-summary-flat.pdf

# Life sciences study design

All studies must disclose on these points even when the disclosure is negative.

| | |
|---|---|
| Sample size | No sample size calculation was performed. For the inference of the phylogenetic tree aminoacid sequences of citrate synthase genes from cyanobacteria were collected from publicly deposited sequences. Sequences were selected in order to represent the diversity of the phylum and to follow the species phylogeny of cyanobacteria as reported in previous publications. For biochemical assays and growth curves 3 replicates were carried out to be able to calculate a standard deviation and keep the sample sizes experimentally manageable. For all Michaelis-Menten plots for the kinetic characterization of enzymes three independent experiments were performed on different days and set up with new substrate preparations. The measurements for the different substrate concentrations were done in three technical replicates for each of those three experiments. For growth curves and survival assays of S. elongatus strains as well as for metabolite extraction three biological replicates (i.e. independent cultures) in shaking flasks were used. |
| Data exclusions | For the inference of the phylogenetic tree of citrate synthases in cyanobacteria Prochlorothrix hollandica could not be recovered robustly as sister group to Synechococcus elongatus PCC 7942. Since this phylogenetic relationship is otherwise well established in the literature we excluded the sequence from the data set. We did use this sequence later to produce a constrained tree, with P. hollandica at the correct position, to verify the inference regarding the evolution of the fractal assembly (s. Extended Data Figure 8b). |
| Replication | Experiments were usually prepared in three biological replicates to ensure robustness of the conclusions. The exact number of replicates is always mentioned in the figure legend. |
| Randomization | NA |
| Blinding | NA |

# Reporting for specific materials, systems and methods

We require information from authors about some types of materials, experimental systems and methods used in many studies. Here, indicate whether each material, system or method listed is relevant to your study. If you are not sure if a list item applies to your research, read the appropriate section before selecting a response.

## Materials & experimental systems

| n/a | Involved in the study |
|-----|----------------------|
| ☒ ☐ | Antibodies |
| ☒ ☐ | Eukaryotic cell lines |
| ☒ ☐ | Palaeontology and archaeology |
| ☒ ☐ | Animals and other organisms |
| ☒ ☐ | Clinical data |
| ☒ ☐ | Dual use research of concern |
| ☒ ☐ | Plants |

## Methods

| n/a | Involved in the study |
|-----|----------------------|
| ☒ ☐ | ChIP-seq |
| ☒ ☐ | Flow cytometry |
| ☒ ☐ | MRI-based neuroimaging |

## Plants

Seed stocks
*Report on the source of all seed stocks or other plant material used. If applicable, state the seed stock centre and catalogue number. If plant specimens were collected from the field, describe the collection location, date and sampling procedures.*

Novel plant genotypes
*Describe the methods by which all novel plant genotypes were produced. This includes those generated by transgenic approaches, gene editing, chemical/radiation-based mutagenesis and hybridization. For transgenic lines, describe the transformation method, the number of independent lines analyzed and the generation upon which experiments were performed. For gene-edited lines, describe the editor used, the endogenous sequence targeted for editing, the targeting guide RNA sequence (if applicable) and how the editor was applied.*

Authentication
*Describe any authentication procedures for each seed stock used or novel genotype generated. Describe any experiments used to assess the effect of a mutation and, where applicable, how potential secondary effects (e.g. second site T-DNA insertions, mosiacism, off-target gene editing) were examined.*

