## [Peer Review File · Nature]

Manuscript Title: Emergence of fractal geometries in the evolution of a metabolic enzyme

Reviewer Comments & Author Rebuttals

Reviewer Reports on the Initial Version:

Referees' comments:

Referee #1 (Remarks to the Author):

In this report, F. L. Sendker and coworkers report that the citrate synthase (CS) from the cyanobacterium *S. elongatus*, a natural protein, self-assembles into Sierpiński triangles. To understand the formation mechanism behind, the authors have experimentally used cryo-electron microscopy to reveal how the fractal assembles from a hexameric building block. They have also revealed that the formation of the fractal complexes is stimulus responsive and can strikingly regulate the enzymatic activity in vitro. Very interestingly, the authors have applied ancestral sequence reconstruction to retrace the evolution of the CS fractal from non-fractal precursors. While being quite attractive and well written, the manuscript has to undergo major revisions before consideration of its possible acceptance for publication in Nature. They lie in actually two major aspects: one is the formation mechanism of the CS Sierpiński triangles, and the other is the origin of the enzymatic activity in vitro of the assembly. Specifically, the main issues are listed below.

1) According to Figure 1, there are at least two cyanobacterial species *S. elongatus* PCC 7942 (Monomer mass = 44.3 kDa) and *Synechocystis* sp. PCC 6803 (Monomer mass = 45.9 kDa) as measured by mass photometry. This means that the starting building blocks are different in size, and hence are quite difficult to assemble into ordered Sierpiński triangles. It seems that the PCC 7942 monomer stands out to the assembling building block to form the Sierpiński triangles. Why? It's previously substantiated (Refs. 10 through 15 in the manuscript) that all assembling building blocks should be identical to form the self-assembled Sierpiński triangles. The authors should elaborate on why they could flexibly assemble different building blocks into the Sierpiński triangles that usually require stringent molecular structures.

2) Moreover, the authors have skillfully applied cryo-electron microscopy to analyze the structures of the fractal assembles formed from a hexameric building block. It's stated in the text that "Only two conservative substitutions occurred at interfaces along this interval: k8R, which is located in the fractal interface and potentially allowed a more stable hydrogen bonding interaction with the backbone of the opposing monomer (Fig. 4e), and y80F, which is located in the older interface connecting dimers into hexamers. In SeCS, F80 engages in a cation- π interaction across the hexamer interface (Fig. 4f)." Is there any other solid experimental evidence to clearly support that a hydrogen bonding interaction with the backbone of the opposing monomer and a cation- π interaction across the hexamer interface are indeed established in the assembled structures? Are the interaction strengths strong and preferential enough to hold the building blocks nearly planarly rather than three-dimensionally?

3) According to the observed structures of the formed CS Sierpiński triangles (Figure 1b), three 6mers

(level 0) can assemble into the 18mers (level 1), and three 18mers, into the 54mers (level 2). During the formation of the 54mers (level 2), the 18mers could assemble into the non-fractal 36mers (Pascal's triangle-like 36mer complexes in Figure 1c). The non-fractal 36mers can be envisioned as three dimers fill inside the large peripheral triangle consisting of 17 dimers. Based on such an observation, one would anticipate that some dimers should be able to attach to the large peripheral triangle from outside, which is actually missing in the proposed models provided in Extended Data Fig. 3 aI through aIII. Why?

4) For the formation of the large and complex Sierpiński triangles, it seems to be quite important to maintain a subtle balance between the connection specificity and flexibility between two 6mers. Does this compromise have anything to do with the apparent physiological pKa of the assembly?

Referee #2 (Remarks to the Author):

In the manuscript "Emergence of fractal geometries in the evolution of a metabolic enzyme," Sendker and colleagues assess the structure, enzyme activity, evolutionary history, and physiological relevance of fractal assemblies formed by the citrate synthase from the cyanobacterium *S. elongatus*. The authors show that the enzyme self-assembles into trimers of hexamers (18mers) and trimers of 18mers (54mers) in a fractal geometry known as Sierpinski triangles. Using a variety of structural techniques, the authors define the interaction interfaces that make the fractal assembly possible, uncovering protein conformational changes that need to occur for assembly. The authors characterize the effects of assembly on activity, finding that hexamers are the active assembly and that higher order assembly, specifically 18mers, slows catalytic activity; however, these effects are not reproduced in vivo under conditions thought to induce 18mer formation. The authors then use phylogenetic analyses to reconstruct the ancestral sequences of citrate synthases in cyanobacteria and characterize both a subset of the modern proteins and the predicted ancestral node proteins. They find that fractal formation by citrate synthase from *S. elongatus* requires very few substitutions to the ancestral proteins, suggesting that fractal formation may be an accident of evolution or a molecular spandrel.

To my knowledge, no other work demonstrating naturally-occurring, regular protein fractals has been published, and the assemblies are quite striking. Overall the manuscript is clear and concise and the data supports the conclusions the authors have drawn. The authors have used appropriate and valid techniques from a number of disciplines to address their questions and the data is of good quality. Points of concern and opportunities for manuscript improvement are summarized below:

Major remarks

1. The authors conclude that fractal assembly is likely not regulating protein activity in vivo and they characterize the assembly as a possible "an accident of history" (pg. 11, line 17-19) rather than serving a current physiological role. This seems like an important finding and should be added to the abstract.
2. The authors discuss how the requirements for a Sierpinski triangle may be difficult to evolve within commonly found protein-protein interfaces (pg 3, line 16-18 and pg 6, line 1-10). However, in the final paragraph (pg 14, line 7-12), the authors suggest transitions in self-assembly may be more common than currently realized. It was unclear to me if characteristics of SeCS make it uniquely possible for it to form

a Sierpinski triangle, or others may exist but the field has not identified additional instances of regular fractal formation. D3 symmetry and protomer flexibility (pg 9, line 15-17) are found in other proteins. Do the authors think that other flexible D3-symmetric proteins could form Sierpinski triangles, or that SeCS and/or citrate synthases have unique properties that allow for fractal formation? This should be clarified.

3. The authors link formation of the 18mer to a decrease in enzyme activity, and suggest the rotation required for fractal assembly is incompatible with substrate binding or catalysis (pg 10, line 12-18). This argument would be strengthened if the authors briefly mention how this rotation affects active and/or allosteric site formation or known structural requirements for catalysis, especially as citrate is a product of the reaction and may induce a conformation different from substrate-bound citrate synthase.

4. In Figure 1b, a hexamer from an 18mer is used to represent the “6mer – level 0” in the series because the 6mer shows preferred orientation for an orthogonal view in negative stain. This is clearly stated in the legend, but the figure panel is more ambiguous. The black-boxed portion of that panel has a different pixel dimension from the rest of the image containing the scale bar. I suggest the authors box the inset to the same size as the 18mer/54mer (which would then include a bit of the 18mer) and put a dotted line around the hexamer to make the use of a hexamer from an 18mer visually clearer.

5. This manuscript dovetails with work from Emmanuel Levy’s group, as Levy and colleagues have explored the evolution of large assemblies from oligomers (<https://doi.org/10.1038/nature23320>) and how evolution can also shield proteins from supramolecular assembly (<https://doi.org/10.1073/pnas.210117119>). Levy and colleagues have shown how a very small number of amino acid changes (as few as a single residue mutation) can alter a protein oligomer’s ability or inability to assemble into larger order structures and should be mentioned within this manuscript.

Minor Remarks

6. While the focus of the manuscript is the higher order assemblies formed from SeCS, the authors refer to the protomers and dimers that make up the hexamer and higher assemblies within their discussions. Protomers versus dimers are difficult to parse in Figure 2b-d, and the color scheme of Figure 2b makes it difficult identify protomers and to distinguish dimers within the hexamer for some of the volumes and the models. A more unified color scheme across this figure as well as clearer identification of protomers and dimers would clarify the findings.

7. For clarity, Figure 3b should be described as a ratio of kinetic measurements within the figure legend.

8. It is unclear to me why a defined 120° angle in a protein monomer should be required (pg 9, line 14), as this geometry might be introduced by oligomerization or conformational variability within the protein. Additionally, C3 and D3 symmetric protein assemblies contain 120° angles, so cells have found a way to create this geometry (pg 4, line 1-3). Unless I am mistaken, the authors may want to clarify or emphasize that these requirements are derived from small molecules and briefly discuss the way in which these requirements apply to proteins.

9. The authors generate a protein model of a 54mer (Extended Data Figure 2), which requires a 22° rotation of a corner dimer to be flat like those observed in negative stain. Does this rotation induce any clashes that would make it physiologically unlikely? If so, are there other ways the protein can flex or

rotate to accommodate the interactions formed between the corner dimers of the 18mers that make up the 54mer? This should be mentioned within the discussion of the 54mer model. (Very minor note: I think the color scheme may be switched for a set of the 18mer “corner” dimers in the 54mer model in Extended Data Figure 2d.)

10. The authors use amino acid nomenclature (for example: L18q) with mixed upper and lower case residue identifiers. It is unclear why this is the case until late in the manuscript, when the nomenclature is defined in relation to the ancestral genes. Is there a way for the authors to define this usage the first time it appears?

11. Because authors discuss specific amino acid contacts within the interacting interfaces identified from their cryoEM structures and draw conclusions from those contacts, I suggest moving the image containing the volume of those residues (Supp. Figure 2i, or an updated equivalent) from the Supplementary Figures into Extended Data Figures (or ideally to the main text Figure) for easier evaluation by readers. Additionally, as the protomers of each hexamer within the 18mer are not equivalent, the authors should identify which protomer volume they are showing in Supp. Figure 2h.

12. The Supplementary Figures and Tables are not referenced in the manuscript (main text, figure legends, or Extended Data). These should be referenced where relevant.

13. The authors should identify the software used for generating images panels, graphs, and statistical calculations either within the specific method description or as a separate method section for data analysis and presentation (though the software is included in the Reporting Summary).

Thank you for the opportunity to read this manuscript, and best wishes for this and future work.

Referee #3 (Remarks to the Author):

The study by Sendker et al. presents a most fascinating assembly phenomenon as applied at the molecular level for an individual protein. The study demonstrates elegantly that this phenomenon can be reconstructed evolutionarily. The study has broad implications for our understanding of oligomeric protein assemblies and protein evolution, and the regulation of enzymatic activity.

Please find below input for the authors' consideration.

The last sentence of the abstract referring to the regulatability of the protein assemblies is rather contradictory to subsequent statements in the manuscript, e.g. “Even though the assembly has many hallmarks of being regulatory (catalytic differences between stoichiometries, responsiveness to physiological conditions), it is apparently not important to fitness, though we cannot rule out that it might be under natural conditions.”

The authors state:

“While characterizing new types of quaternary structures in this family, we discovered that the CS from the cyanobacterium *Synechococcus elongatus* PCC 7942 (SeCS and *S. elongatus* hereafter) forms an

unusual assembly.”

Please qualify and briefly describe the strategy and criteria for the claimed characterization of new types of quaternary structures, and how this connects to the topic of the manuscript. Furthermore, it is unclear why bacteria citrate synthases were targeted in this campaign. There are many protein families that could serve as candidates in a such an undertaking.

How does the concentration of protein samples used for the mass photometry measurements and SAXS compared with those used for the initial characterization thereof by negative-stain EM and the subsequent structure determination by cryo-EM? It would be useful to state the used concentrations using the same concentration units throughout the manuscript.

This is because the observed Sierpinski triangle assemblies are concentration dependent.

For instance, this reviewer calculates that the protein sample used for negative-stain EM was at ~300 nM, which is 6-times higher than the concentration used for negative-stain EM.

Furthermore, the authors do not disclose the concentration of sample used for cryo-EM whatsoever. This needs to be rectified and expressed in terms of consistent unit designations used for all other methods in the manuscript.

Fig. 1 panels b&c: How populated are the presented 2D classes from negative-stain EM analysis?

What is the projected size of the fractals beyond the observed 54-mers?

What does the micrograph presented in Ext. Data Fig. 1g suggest about the possible n-mer size of the largest partial Sierpinski triangle shown? How does this relate to the other findings reported in the manuscript?

The functional and enzymatic data presented in Figure 3a at different substrate and product concentrations are puzzling to this reviewer. This is because at the SeCS concentration used (25 nM) in these experiments, the determined fraction of 6mer vs 18 mer does not appear to follow the data presented in Figure 1a, which is based on SeCS at 50 nM.

What are the expected structural consequences of mutating H369 to Arginine in SeCS? The authors need to show that introduction of the H369R mutation does not structurally compromise SeCS before making connections to the pH dependence of SeCS enzymatic activity.

What is the rationale of substituting H369 with an arginine in the first place.

What does the data suggest about the pKa of H369 in WT SeCS? What do calculations or other experimental data suggest about the pKa of H369?

The authors ponder about the evolutionary lability of protein self-assembly en route to stable protein fractals.

The authors do not discuss any thermodynamic considerations, which in principle could provide the appropriate biophysical framework to explain such phenomena.

Extended Data Table 1: The units for Map sharpening B factor and B factors should be corrected to square Å. Also there seems to be some typographical issues with the values entered for B factors and the values in the next line of the table corresponding to Protein residues.

Extended Data Table 2:

Rmerge is a flawed crystallographic data indicator. Please report Rmeas and/or Rpim.

I / σI needs to be corrected to I/σI

Units of square Å are missing for B-factors

Referee #4 (Remarks to the Author):

The authors presented an interesting case of protein oligomerization example. They found that one of extant CS protein demonstrated high oligomerization state with fractal formation. It is indeed fascinating see how such systematic oligomeric state can be observed in the molecular level. The paper is generally written very well and easy to digest. The biophysical experiments are detailed and well executed and presented to show how Sierpinski assembly states are formed. Technical details of experimental procedures are well explained.

The major issue I have on this paper is that the fractal state does not have any evolutionary meaning as far as the authors presented. There are not other close homologous CS exhibit similar fractal state. This suggest that it is likely that there is no selection occurred to the cyanobacterium *Synechococcus elongatus*. Also the authors demonstrated that there is no observed effects fitness. The activity difference is only two-fold and it is not clear this can substantial in undiscovered natural conditions.

Thus, at the end, it seems that this paper showed an interesting oligomeric state, which is unlikely related to a large functional consequence. Frankly, the lack of biological relevance turn my enthusiasm off, and I believe that this work might be better suited for more specialized journals.

Author Rebuttals to Initial Comments:

Point by point answers:

Reviewer 1

We are grateful for this reviewer's constructive feedback. Based on their suggestions we performed several additional experiments that helped us clarify the assembly mechanism of this fascinating protein.

1) According to Figure 1, there are at least two cyanobacterial species *S. elongatus* PCC 7942 (Monomer mass = 44.3 kDa) and *Synechocystis* sp. PCC 6803 (Monomer mass = 45.9 kDa) as measured by mass photometry. This means that the starting building blocks are different in size, and hence are quite difficult to assemble into ordered Sierpiński triangles. It seems that the PCC 7942 monomer stands out to the assembling building block to form the Sierpiński triangles. Why? It's previously substantiated (Refs. 10 through 15 in the manuscript) that all assembling building blocks should be identical to form the self-assembled Sierpiński triangles. The authors should elaborate on why they could flexibly assemble different building blocks into the Sierpiński triangles that usually require stringent molecular structures.

This is a misunderstanding. We have never shown or observed assembly into Sierpiński triangles comprising more than one building block. The CS from *S. elongatus* PCC 7942 is the one that can assemble into Sierpiński triangles. Its complexes are built exclusively from identical building blocks and no mixtures. In contrast, the CS from *Synechocystis* sp. PCC 6803 does not form Sierpiński triangles and can only assemble into hexameric complexes and no larger structures. We only show it in Fig. 1a as a reference to compare a "normal" CS assembly, which we find in most other CS from Cyanobacteria (Extended Data Fig. 8) and many other bacteria, to the unusual complexes we found for the CS from *S. elongatus* PCC 7942. We also wanted to highlight that the subcomplexes (2mer, 6mer) of the fractal 18mers are the same as for "normal" CS proteins. We adapted Fig.1a and its description to prevent this confusion.

2) Moreover, the authors have skillfully applied cryo-electron microscopy to analyze the structures of the fractal assemblies formed from a hexameric building block. It's stated in the text that "Only two conservative substitutions occurred at interfaces along this interval: k8R, which is located in the fractal interface and potentially allowed a more stable hydrogen bonding interaction with the backbone of the opposing monomer (Fig. 4e), and y80F, which is located in the older interface connecting dimers into hexamers. In SeCS, F80 engages in a cation- π interaction across the hexamer interface (Fig. 4f)." Is there any other solid experimental evidence to clearly support that a hydrogen bonding interaction with the backbone of the opposing monomer and a cation- π interaction across the hexamer interface are indeed established in the assembled structures? Are the interaction strengths strong and preferential enough to hold the building blocks nearly planarly rather than three-dimensionally?

The cryo-EM density indicated that the two residues are taking part in the interaction of the 18mer interface (8R) and the hexamer interface (80F); local density of the residues is shown in Extended Data Fig. 10d+e. Both interactions are detected by Pymol's in-built cation- π and hydrogen-bond detection routines. We do acknowledge, however, that our resolution is poor, so these are structural hypotheses, rather than proven facts. We have amended the relevant statements to indicate this (p. 14, lines 14-17)

What remains undoubtedly true is the biochemical effects of substitutions at these sites. Introducing these substitutions together into ancB increased the abundance of 18mer complexes at nanomolar concentrations (now Fig. 5g). To additionally prove their contribution, we now also performed the opposite experiment and reversed the substitutions in the wildtype SeCS to the ancestral state found in ancB: R8k and F80y. Both changes reduced the abundance of 18mer complexes at nanomolar concentrations, which further supports the inference that these two interactions are important to hold the hexamers in the planar 18meric complexes. These measurements are added to Extended Data Fig. 10f (see below) and discussed in the manuscript (p. 14, lines 19-20).

3) According to the observed structures of the formed CS Sierpiński triangles (Figure 1b), three 6mers (level 0) can assemble into the 18mers (level 1), and three 18mers, into the 54mers (level

2). During the formation of the 54mers (level 2), the 18mers could assemble into the non-fractal 36mers (Pascal's triangle-like 36mer complexes in Figure 1c). The non-fractal 36mers can be envisioned as three dimers fill inside the large peripheral triangle consisting of 17 dimers. Based on such an observation, one would anticipate that some dimers should be able to attach to the large peripheral triangle from outside, which is actually missing in the proposed models provided in Extended Data Fig. 3 aI through aIII. Why?

This is an interesting point. Our new structure of a 54mer shows that even in this geometry, all edges are passivated. Once a Sierpiński triangle has assembled, it is impossible to add subunits to anything but the corners (Extended Data Fig. 2e-f). As the reviewer points out, the 36mer seems to violate this rule. But we do not believe these would form by filling a pre-existing triangle with three dimers. This would require first the formation of a 30mer, in which three of its subcomplexes are missing dimers. We have no evidence that structures like this can form. Rather, we believe 36mers assemble separately, without going through a Sierpiński intermediate.

Still, the 36mers must be mainly built from the same interface as the 18mers (and all larger fractal forms) by connecting 6 of them in a triangular shape. We know this through two lines of evidence which we also discuss in the manuscript (p. 8, line 21 – p. 9, line 13): When we disrupt this interface by introduction of the steric clash L18Q, we only observe hexamers (Fig. 5d). Even at extremely high concentrations of up to 250 μM the sensitive SAXS measurements only show the formation of hexamers and no other larger species (Fig. 1f). We also rule out the possibility of an additional interface that overlaps with the fractal interface by a variant that has an intact fractal interface but a weakened hexamer interface and mostly forms tetramers (D147A, Extended Data Fig. 3b-e). We now also created an additional combination variant (L18Q+D147A) that further shows that all larger oligomers that were detected in the native MS measurements of the D147A variant are formed because the hexameric interface was not fully abolished and not due to an additional interface (Extended Data Fig. 3f-i).

Therefore, we know that 36mers are also connected via the fractal interface which can only support a two-way connection. It further passivates the created edge of the formed triangles in the same way as the 18mers and additional dimers cannot be added to the outside of the triangle without causing a steric clash. These are the reasons why we did not include the proposed variants as models for the 36mer.

4) For the formation of the large and complex Sierpiński triangles, it seems to be quite important to maintain a subtle balance between the connection specificity and flexibility between two 6mers. Does this compromise have anything to do with the apparent physiological pKa of the assembly?

The reviewer is exactly right. Our new 54mer structure proves this point beautifully: The interaction between 18mers uses the same surface, but makes the interaction at a shallower dihedral angle across dimers to allow the larger triangle to close (Fig. 3c). The ability to make larger fractals therefore evidently relies on this angular flexibility. We now emphasize the importance of this flexibility in the text.

We do not, however, think that this is directly connected to the pKa of the assembly, as we see this behaviour also in our mutant protein, in which the interfacial histidine has been replaced by an arginine.

Reviewer 2

We are grateful for the reviewer's very positive assessment of our work. Based on their suggestions, we have made numerous adjustments to the text and figures, which we feel greatly improved the clarity of the paper.

MAIN

1) The authors conclude that fractal assembly is likely not regulating protein activity in vivo and they characterize the assembly as a possible "an accident of history" (pg. 11, line 17-19) rather than serving a current physiological role. This seems like an important finding and should be added to the abstract.

We agree and adapted the abstract p. 2, lines 17-21

"We show that although the formation of fractal complexes is stimulus responsive and can regulate the enzymatic activity in vitro, it may not serve a physiological function in vivo. We retrace how the citrate synthase fractal evolved from non-fractal precursors using ancestral sequence reconstruction suggesting it may have emerged as a harmless evolutionary accident."

2) The authors discuss how the requirements for a Sierpiński triangle may be difficult to evolve within commonly found protein-protein interfaces (pg 3, line 16-18 and pg 6, line 1-10). However, in the final paragraph (pg 14, line 7-12), the authors suggest transitions in self-assembly may be more common than currently realized. It was unclear to me if characteristics of SeCS make it uniquely possible for it to form a Sierpiński triangle, or others may exist but the field has not identified additional instances of regular fractal formation. D3 symmetry and protomer flexibility (pg 9, line 15-17) are found in other proteins. Do the authors think that other flexible D3-symmetric proteins could form Sierpiński triangles, or that SeCS and/or citrate synthases have unique properties that allow for fractal formation? This should be clarified.

The requirements for the assembly are only difficult to evolve in the sense that they require types of interaction that are as of yet underrepresented in the PDB, which is still dominated by crystal structures that are easier to obtain for assemblies that correspond to point group symmetries. In fact, our reconstructions show that this assembly was shockingly easy to evolve. We agree with the reviewer that there is nothing special about this protein or its evolutionary precursors, which are a garden-variety D3 homo-oligomers. It is therefore quite plausible that other families can or have evolved the ability to form Sierpiński triangles. Until quite recently, it may have simply been too difficult to discover them. We have added these points to our discussion, as the reviewer suggested (p. 9 line 23 – p. 10 line 5).

More broadly, our results imply that previous catalogues of what complexes are feasible are likely too restrictive, because they for example exclude non-bijective complexes in which not all subunits participate equally. If Sierpiński triangles are possible when these restrictions are lifted, a whole host of so-far undiscovered 2D and 3D protein materials may also be achievable. We are currently initiating efforts to discover such assemblies.

3) The authors link formation of the 18mer to a decrease in enzyme activity, and suggest the rotation required for fractal assembly is incompatible with substrate binding or catalysis (pg 10, line 12-18). This argument would be strengthened if the authors briefly mention how this rotation affects active and/or allosteric site formation or known structural requirements for catalysis, especially as citrate is a product of the reaction and may induce a conformation different from substrate-bound citrate synthase.

Citrate synthases have been extensively structurally studied and it has been known for a long time that they exist in so-called “open” and “closed”-forms. The closed form is found in structures that are bound to ligands – oxaloacetate (4TVM), citrate (6ABX) or combinations with acetyl-CoA or its non-hydrolysable analogues (2H12, 1IXE, 4CTS). It is also known that this closed form is the conformation in which catalysis takes place while the open form allows for substrate binding and product release (PMID: **2043640**). The conformational change from open to closed form is described as a rigid body rotation by mainly the small subdomain of the monomer, which corresponds to the alpha-helices close to the substrate binding pocket (PMID: 27493854, PMID: 3013232). This is exactly what we observe when we compare the structure of the hexameric Δ 2-6 SeCS, which corresponds to the open form, and the citrate-bound SeCS, which corresponds to the closed form (added now in Extended Data Fig. 6c, see below). As shown in the manuscript, SeCS undergoes a rigid-body rotation in the opposite direction when forming the 18meric complexes (Fig. 3b). Therefore, the structural rearrangement from the fractal form into the closed form which allows for catalysis is much larger. This likely imposes a higher energetic barrier which could explain the decrease in enzyme activity.

We agree with the reviewer that this has not been sufficiently put into context in the manuscript and we have now added this information (p. 11, lines 6-13)

4) In Figure 1b, a hexamer from an 18mer is used to represent the “6mer – level 0” in the series because the 6mer shows preferred orientation for an orthogonal view in negative stain. This is clearly stated in the legend, but the figure panel is more ambiguous. The black-boxed portion of that panel has a different pixel dimension from the rest of the image containing the scale bar. I suggest the authors box the inset to the same size as the 18mer/54mer (which would then include a bit of the 18mer) and put a dotted line around the hexamer to make the use of a hexamer from an 18mer visually clearer.

We agree with the reviewer and adapted the figure (now Fig. 1c).

5) This manuscript dovetails with work from Emmanuel Levy’s group, as Levy and colleagues have explored the evolution of large assemblies from oligomers (<https://doi.org/10.1038/nature23320>) and how evolution can also shield proteins from supramolecular assembly (<https://doi.org/10.1073/pnas.2101117119>). Levy and colleagues have shown how a very small number of amino acid changes (as few as a single residue mutation) can alter a protein oligomer’s ability or inability to assemble into larger order structures and should be mentioned within this manuscript.

The Levy group's work is an important inspiration for our studies that we should indeed have cited. We have added it to the discussion together with other studies that explored changes in oligomeric state via individual substitutions (p. 14, line 22 – p. 14, line 3).

“This is consistent with previous studies that have shown that individual substitutions can substantially shift occupancy of oligomeric states or induce supramolecular assembly (PMID: 32461643, **PMID: 28783726**, PMID: 18187656, PMID: 36382881)”

MINOR

6) While the focus of the manuscript is the higher order assemblies formed from SeCS, the authors refer to the protomers and dimers that make up the hexamer and higher assemblies within their discussions. Protomers versus dimers are difficult to parse in Figure 2b-d, and the color scheme of Figure 2b makes it difficult identify protomers and to distinguish dimers within the hexamer for some of the volumes and the models. A more unified color scheme across this figure as well as clearer identification of protomers and dimers would clarify the findings.

We agree with the reviewer and adapted the color scheme and show each protomer in Fig. 2b and 3a-c.

7) For clarity, Figure 3b should be described as a ratio of kinetic measurements within the figure legend.

We agree that the current description is not precise and changed it accordingly.

Now Fig 4b, p. 24, lines 5-7: “Kinetic measurements of SeCS and a hexameric variant (L18q) at different substrate concentrations. Displayed is the ratio of turnover numbers (SeCS/L18q, error bars = SD, n = 3).”

8) It is unclear to me why a defined 120° angle in a protein monomer should be required (pg 9, line 14), as this geometry might be introduced by oligomerization or conformational variability within the protein. Additionally, C3 and D3 symmetric protein assemblies contain 120° angles, so cells have found a way to create this geometry (pg 4, line 1-3). Unless I am mistaken, the authors may want to clarify or emphasize that these requirements are derived from small molecules and briefly discuss the way in which these requirements apply to proteins.

The reviewer is right that 120° angles occur in C3 and D3 interfaces. This is, however, not enough to produce a Sierpiński triangle, as we show in Figure 2a, because such interfaces alone cannot produce passivated edges. In our case the problem is solved through the pseudosymmetrical interface we discovered between dimers. In synthetic fractals the 120° angle within the monomer solves the same problem: A single monomer bridges the different sub-triangles and its backbone ensures that this occurs at the right angle (PMID: 25901816). Using a kinked monomer to connect sub-triangles in this way ensures that the edges are passivated. If sub-triangles were connected via another C3 interface, this would cause the same problems that we detail in Figure 2a: It would allow a third subunit to associate into the edge, resulting in a crystalline lattice. We added this distinction to the main text to clarify it (p. 3, lines 20-23)

9) The authors generate a protein model of a 54mer (Extended Data Figure 2), which requires a 22° rotation of a corner dimer to be flat like those observed in negative stain. Does this rotation induce any clashes that would make it physiologically unlikely? If so, are there other ways the protein can flex or rotate to accommodate the interactions formed between the corner dimers of the 18mers that make up the 54mer? This should be mentioned within the discussion of the 54mer model. (Very minor note: I think the color scheme may be switched for a set of the 18mer “corner” dimers in the 54mer model in Extended Data Figure 2d.)

Our new structure of a 54mer has clarified this issue. Rather than rotating the dimer at the corners of the 18mer more extremely, the interaction between 18mers is made at shallower dihedral angle than the interaction between 6mers within an 18mer (now Fig. 3c). The resolution of our density is unfortunately not sufficient to resolve how this change in angle affects the contacts in the interface. We therefore prefer not to theorize about why this interaction seems to form less readily than that which holds 6mers into 18mers.

There was indeed a switch in the color scheme in Extended data Fig. 2d (now 2g) – good catch! We corrected the figure.

10) The authors use amino acid nomenclature (for example: L18q) with mixed upper and lower case residue identifiers. It is unclear why this is the case until late in the manuscript, when the nomenclature is defined in relation to the ancestral genes. Is there a way for the authors to define this usage the first time it appears?

The reviewer is correct that we did not introduce the nomenclature the first time we use it in the manuscript. We now only use this nomenclature in the section about the evolution of the fractal, where we now explicitly define it (p. 13, lines 20-21).

11) Because authors discuss specific amino acid contacts within the interacting interfaces identified from their cryoEM structures and draw conclusions from those contacts, I suggest moving the image containing the volume of those residues (Supp. Figure 2i, or an updated

equivalent) from the Supplementary Figures into Extended Data Figures (or ideally to the main text Figure) for easier evaluation by readers. Additionally, as the protomers of each hexamer within the 18mer are not equivalent, the authors should identify which protomer volume they are showing in Supp. Figure 2h.

The figures with close-ups of the cryo-EM density of important amino acids have been updated and moved to Extended Data Fig. 2b and 10b+c-d. The reviewer is also correct that we should identify the protomer volume from the structure – we have done that now and updated Supplementary Fig. 3h.

12) The Supplementary Figures and Tables are not referenced in the manuscript (main text, figure legends, or Extended Data). These should be referenced where relevant.

Agreed and adapted.

13) The authors should identify the software used for generating images panels, graphs, and statistical calculations either within the specific method description or as a separate method section for data analysis and presentation (though the software is included in the Reporting Summary).

Agreed and adapted.

Reviewer 3

We are very grateful for the reviewer's critical and constructive assessment of our work. Their suggestions prompted us to eventually solve the structure of a 54mer – the second level on the Sierpiński fractal, which greatly improved our understanding and the impact of the work. We have also made numerous adjustments to our text and figures based on their suggestions, as detailed below.

1) The last sentence of the abstract referring to the regulatability of the protein assemblies is rather contradictory to subsequent statements in the manuscript, e.g. "Even though the assembly has many hallmarks of being regulatory (catalytic differences between stoichiometries, responsiveness to physiological conditions), it is apparently not important to fitness, though we cannot rule out that it might be under natural conditions."

We modified the abstract to include the notion that the fractal might have evolved as an "accident of history" – see also our response to Reviewer 2, point 1 (p. 1, lines 17-21). But we

want to highlight that the assembly itself is regulatable (responsive to pH and high substrate concentrations) but that this feature is apparently not exploited by the host organism at least not under the conditions we tested.

*2) The authors state: "While characterizing new types of quaternary structures in this family, we discovered that the CS from the cyanobacterium *Synechococcus elongatus* PCC 7942 (SeCS and *S. elongatus* hereafter) forms an unusual assembly." Please qualify and briefly describe the strategy and criteria for the claimed characterization of new types of quaternary structures, and how this connects to the topic of the manuscript. Furthermore, it is unclear why bacteria citrate synthases were targeted in this campaign. There are many protein families that could serve as candidates in a such an undertaking.*

The truth is there was no strategy and also no intention to look for a fractal protein. We cloned and purified the CS from *S. elongatus* for a project to compare it to different bacterial CS enzymes and noticed via MP that this specific protein forms unusually large complexes. From there on we kept investigating. But to prevent any confusion we deleted the half sentence (p. 4, line 12-13).

3) How does the concentration of protein samples used for the mass photometry measurements and SAXS compared with those used for the initial characterization thereof by negative-stain EM and the subsequent structure determination by cryo-EM? It would be useful to state the used concentrations using the same concentration units throughout the manuscript. This is because the observed Sierpiński triangle assemblies are concentration dependent. For instance, this reviewer calculates that the protein sample used for negative-stain EM was at ~300 nM, which is 6-times higher than the concentration used for negative-stain EM.

We agree this was confusing and now show all concentrations as either micro- or nanomolar. The mass photometry measurements were done at 50 nM. Negative stain EM was done at 450 nM (approx. 9x higher than MP concentration). SAXS measurements were done between 2.5 μ M (approx. 5.5x higher than the negative stain concentration) and 250 μ M (approx. 550x higher than the negative stain concentration). The concentration of the cryo-EM sample was 22.5 μ M (approx. 50x higher than the negative stain concentration).

4)Furthermore, the authors do not disclose the concentration of sample used for cryo-EM whatsoever. This needs to be rectified and expressed in terms of consistent unit designations used for all other methods in the manuscript.

This was an oversight. The protein concentration for cryo-EM was 22.5 μ M and is added to the methods part of the manuscript. We also adapted the manuscript to state protein concentrations consistently in either micro- or nanomolar units (see above).

5) *Fig. 1 panels b&c: How populated are the presented 2D classes from negative-stain EM analysis?*

As mentioned in the methods part of the manuscript it is very difficult to specify the prevalence of the individual particles from the negative stain analysis because of the strong preferential orientation. Particles highly prefer to fall on the edge of triangles rather than on the flat surface and therefore appear as elongated rectangles. The 2D class averages presented in Fig. 1b+c were created from a large set of 500 micrographs in which we specifically sought after larger assemblies. In this data set 36mers (186 particles) and 54mers (200 particles) appeared roughly equally often. The analysis was automated using cisTEM and only included top views of the particles. To get a better idea of how populated these assemblies are compared to 18mers we now collected another dataset of 150 micrographs without a bias towards larger assemblies. We counted all particles by hand for these micrographs and included the assemblies that were laying on their edge and appeared as rectangles. By measuring the edge length, we could assign them to be either a 36mer (30 nm) or 54mers (40 nm). The result of this analysis revealed that at a protein concentration of 450 nM approx. 92.8 % of detected assemblies were 18mers (1773 particles), 3.5 % were identified as 36mers (66 particles) and 3.8% were identified as 54mers (72 particles).

We caution here and in the manuscript is still a relatively rough estimate as identification of the individual assemblies was difficult due to the low resolution and contrast of negative stain EM. Our estimate should therefore be taken with care and by comparison with our SAXS data, which show that large complexes only start being reasonably common above 25 μ M protein concentration (Fig. 1f). We added this new estimate to the manuscript as well (p. 4, lines 21-23).

6) *What is the projected size of the fractals beyond the observed 54-mers?*

The edge length of an 18mer (level 1) is almost exactly 20 nm and always doubles for the next fractal level. This results in an edge length of 40 nm for 54mers (level 2), 80nm for 162mers (level 3) and 160 nm for 486mers (level 4).

7) *What does the micrograph presented in Ext. Data Fig. 1g suggest about the possible n-mer size of the largest partial Sierpiński triangle shown? How does this relate to the other findings reported in the manuscript?*

The large assembly we show in Ext. Data Fig. 1g is not a Sierpiński-triangle but rather reminiscent of a 2D lattice-structure that is enclosed in a triangular shape where the edges violate the lattice-pattern. We created a simple model (added to Extended Data Fig. 1g, see below) which indicates that the assembly contains more than 300 CS subunits. We built this model from hexamers connecting via the same interaction as the 18mer. The grey coloured dimers run into the same

problem we have discussed for 36mers. Three dimers come together but the interface only allows a two-way connection. Therefore, not all interfaces are satisfied here.

To relate this to the other findings in our manuscript: It is our hypothesis that with an increasing number of subunits there are an increasing number of assemblies that are not Sierpiński-triangles but that the protein can assemble into. While Sierpiński-triangles maximize interactions and only have three unsatisfied interactions at their corners (Extended Data Fig. 4a), the number of subunits between two fractal levels increases steeply (18, 54, 162 etc.). Therefore, alternative assembly forms might be favoured to form as well (like the 36mers). Lastly, we want to note that we have observed this type of assembly only once in all our micrographs and the existing information is therefore limited. We included it as it illustrates again that the protein favours forming triangles.

8) The functional and enzymatic data presented in Figure 3a at different substrate and product concentrations are puzzling to this reviewer. This is because at the SeCS concentration used (25 nM) in these experiments, the determined fraction of 6mer vs 18 mer does not appear to follow the data presented in Figure 1a, which is based on SeCS at 50 nM.

With hindsight, our presentation of this data was confusing. The fractions presented in Figure 3a (now Fig. 4a) correspond to CS subunits in the respective oligomeric states. This means a dimer

accounts for 2 subunits and an 18mer accounts for 18 subunits – so 9x more than a dimer. In contrast, the mass photometry measurements e.g. Fig. 1a present a histogram of oligomeric particles where one dimer and one 18mer both account for one particle each. The distribution therefore looks different. To make this more obvious we added a panel to Figure 1 with a bar graph that shows the distribution of subunits within different oligomeric species from the MP measurement of SeCS at 50 nM (Fig. 1b, see below).

Fig. 1. The CS of *S. elongatus* PCC 7942 assembles into Sierpiński triangles (a) Distribution of oligomeric protein complexes of purified CS from two cyanobacterial species *S. elongatus* PCC 7942 (SeCS, monomer mass = 44.3 kDa) and *Synechocystis* sp. PCC 6803 (Monomer mass = 45.9 kDa) measured by mass photometry (MP). Cartoons represent assembly of known CSs. (b) Distribution of SeCS subunits in the different oligomeric complexes corresponding to the MP measurement in (a).

9) *What are the expected structural consequences of mutating H369 to Arginine in SeCS? The authors need to show that introduction of the H369R mutation does not structurally compromise SeCS before making connections to the pH dependence of SeCS enzymatic activity. What is the rationale of substituting H369 with an arginine in the first place.*

We are grateful for this suggestion, because it ultimately allowed us to solve the structure of a 54mer. First to explain why we made this mutant: we observed that fractal complexes are disrupted at higher pH, we wanted to investigate the structural basis for the pH sensitivity. Histidine H369 was an obvious candidate because it is part of the interface that holds together hexamers through a hydrogen bond (Fig. 2c). To experimentally verify this theory, we substituted the histidine with a positively charged but non-titratable arginine. Mass photometry data revealed that the variant forms 18mers and behaves the same as the wildtype at pH 7-7.5, but that it does not dissociate at high pH (Now in Fig. 4d and see response to reviewer 1, point 4). To further prove that introducing H369R does not structurally compromise SeCS we collected negative stain TEM and solved a cryoEM structure for this variant. The data was added to Extended Data Fig. 6d-g (see below).

(d) Negative stain 2D class average of the 18mer formed by H369R SeCS at 450nM. **(e)** Negative stain micrograph showing a 54mer formed by H369R SeCS at 450nM. **(f)** Close up on the interaction between R369 and E6 in the structure H369R SeCS. **(g)** Cryo-EM density of an 18mer from the H369R SeCS variant resolved to 3.52 Å.

As expected, the assembly is virtually unchanged. As predicted, the arginine we introduced appears to form a salt bridge with E6, though bearing in mind the relatively modest resolution of our structure (3.52 Å). To additionally verify that position 369 makes a contact across the interface, we also mutated it to alanine and found the protein to dissociate completely into 6mers (Data added to Extended Data Fig. 2a).

As a happy side-effect of the H369R mutation, the protein appeared to give much better cryoEM grids than the wild-type protein. Specifically, wild-type SeCS aggregated on our cryoEM grids. Together with a strong tendency for preferential orientation, this meant that we saw almost no top views of 54mers on our wild-type grids (even though we had no trouble identifying them on negative stain grids that were prepared at lower concentrations). The H369R mutant aggregated less when plunge frozen, perhaps because it does not have a pH labile interface. Because of the superior quality of the grids, we were able to identify enough top views of 54mers to solve a 5.9 Å structure, which greatly enhanced our understanding of the assembly mechanism (added now in Fig. 3b and 4c, see below).

3**(b)** Cryo-EM density maps of Sierpiński triangle of the 0th, 1st and 2nd fractal level. The 6mer (3.1 Å) was derived from the hexameric Δ2-6 SeCS variant. The 18mer (3.9 Å) was derived from the wildtype SeCS. The 54mer (5.9 Å) was derived from the pH-stabilized variant H369R SeCS.

4**(c)** Dihedral angles between dimers interacting across the fractal interface are depicted for the connection within and between 18mers in a 54mer. Dimers in shown as blue and red, and green and black outlines, respectively.

10) What does the data suggest about the pKa of H369 in WT SeCS? What do calculations or other experimental data suggest about the pKa of H369?

Our language was imprecise with respect to this problem. Our mass photometry experiments on WT SeCS showed that the interface as a whole has a pKa of about 8.5. As described above, we suspected that this pH dependence was mostly caused by histidine 369. We confirmed this suspicion by mutating the histidine to a non-titratable arginine, which abolished the pH sensitivity.

Our manuscript then made the point that the pKa of this histidine apparently already matched the physiological pH change even before the fractal evolved. This was imprecise: we did not measure that pKa directly. Instead, we made use to the phylogenetic age of ancC to make this inference. AncC precedes the evolution of the new interface. It therefore could not have experienced any direct selection pressure for a fractal interface that has approximately the right pKa to match the daily fluctuations of *S. elongatus*. We then created the interface by introducing the substitution q18L into this ancestor. The resulting 18mers were pH sensitive over approximately the same range (Extended Fig. 10g), meaning that the residues at what later became the fractal interface already had the right properties to produce this pKa even before the interaction had evolved. We have clarified this in the text (p. 15, lines 2 - 9)

11) The authors ponder about the evolutionary lability of protein self-assembly en route to stable protein fractals. The authors do not discuss any thermodynamic considerations, which in principle could provide the appropriate biophysical framework to explain such phenomena.

The reviewer is of course right that thermodynamics is the appropriate framework to understand why single mutations can have the effects they do. We referenced this throughout the paper, acknowledging previous work that shows why removing steric repulsion or adding single productive contacts can in fact shift equilibria substantially. We have refrained from fitting explicit thermodynamic models to directly quantify these effects because we have no good way to quantify the abundance of all larger kinds of oligomers at high concentrations. SAXS only gives an ensemble measurement and native mass spectrometry experiments (which in principle could resolve individual complexes) proved infeasible: In extensive trials, the wild-type protein always immediately clogged the nano-spray needles, presumably because some aspect of the nano-spray process induces assembly into very large complexes. For this reason, our discussion of thermodynamics has to remain largely conceptual in this piece of work.

12) Extended Data Table 1: The units for Map sharpening B factor and B factors should be corrected to square Å. Also there seems to be some typographical issues with the values entered for B factors and the values in the next line of the table corresponding to Protein residues.

The table was corrected accordingly (Extended Data Table 1).

13) Extended Data Table 2: Rmerge is a flawed crystallographic data indicator. Please report Rmeas and/or Rpim. I / σI needs to be corrected to I/σI. Units of square Å are missing for B-factors

The units were corrected and the values for Rpim and Rmeas were added to the table (Extended Data Table 2).

Reviewer 4

We appreciate the reviewer's frank feedback and have done more experiments to test for a functional role of the fractal (detailed below) and welcome the opportunity to address their concerns about the evolutionary relevance of our assembly.

The major issue I have on this paper is that the fractal state does not have any evolutionary meaning as far as the authors presented. There are not other close homologous CS exhibit similar fractal state. This suggest that it is likely that there is no selection occurred to the cyanobacterium Synechococcus elongatus. Also the authors demonstrated that there is no observed effects fitness. The activity difference is only two-fold and it is not clear this can substantial in undiscovered natural conditions.

Thus, at the end, it seems that this paper showed an interesting oligomeric state, which is unlikely related to a large functional consequence.

We thank the reviewer for their frank assessment of our work, but we disagree with the conclusion that this discovery has no evolutionary meaning because selection has likely not acted on it. The goal of evolutionary analysis is to discover the ultimate causes of biological forms. Natural selection for some adaptive function is only one of several possible ultimate causes. Other possible explanations are plain chance, historical constraint, or differential mutational accessibility of certain forms under purifying selection. These alternative causes are just as important to discover as selection in order to understand why living organisms have the features they do.

Pan-adaptationism is still the major explanatory framework in biochemistry, which assumes nearly all features of organism at the molecular scale to be highly optimized by selection. We therefore think our study serves as an important counterpoint to this school of thought. In fact, we think it remarkable that something as complex as this assembly apparently could perhaps evolve for no particular reason. It also highlights how important it is to actually test adaptive theories, rather than to accept them based on biochemical plausibility. We are convinced that

this will change the way biochemists view the meaning of many other assemblies that may or may not have obvious adaptive uses.

It remains, however, possible that our original experiments were too crude to detect a potential function. We therefore performed additional *in vivo* experiments to further investigate if there is a fitness effect when *S. elongatus* does not have the ability to form fractal complexes. Nitrogen is often a limiting resource for non-diazotrophic cyanobacteria like *S. elongatus* and the intracellular nitrogen/carbon status is strongly regulated in many cyanobacteria (PMID: 32438704). In *Synechocystis* sp. PCC6803 it has been shown for example that the activity of PEPC, whose main function in cyanobacteria is the anaplerotic synthesis of oxaloacetate to refill the TCA cycle, is linked to the global carbon/nitrogen status and strongly regulated (PMID: 32274833). But this mechanism is not present in *S. elongatus* (PMID: 27911809). We therefore performed recovery experiments under extended nitrogen depletion (chlorosis) to investigate if the formation of fractals regulates TCA-cycle activity to prevent depletion. But we again did not find a difference between the strains that have either WT fractal forming SeCS or the hexameric L18q SeCS (added to the manuscript Fig. 4f, see below).

Lastly, the reviewer notes that a 50% in activity reduction is not enough to plausibly have an effect *in vivo*. This is somewhat misleading: the measurements derive from experiments in which a substantial fraction of monomers still resides in (active) 6mers. The fact that we get such a substantial reduction regardless probably means that 18mers are barely if at all capable of catalysis if they are not allowed to dissociate. All that would be required to turn this into a powerful regulatory mechanism would be some allosteric binder that shifts the equilibrium towards fractals (and would therefore oppose substrate induced dissociation). We currently have no evidence that such a binder exists in *S. elongatus*, but this is a common mechanism for enzyme regulation (for example PMID: 35953658). At the very least, our discovery therefore shows how close an enzyme can come to a regulatory mechanism in just a few mutations - as a result of opposing demands for assembly and catalysis.

4(f) Survival of genetically modified *S. elongatus* strains with either WT or hexameric L18Q-variant of the CS under nitrogen-deficiency for extended periods of time. Serial dilutions of three independent cultures are shown for each time point.

We interpret these findings as implying that under a variety of laboratory conditions, fractal assembly is not important to fitness. It of course remains possible that our experiments are not sensitive enough to detect a very small effect (which in large enough populations would still be visible to natural selection), or that we simply did not test the right conditions. But as we argue above: the absence of a function does not mean this phenomenon has no evolutionary relevance: It clearly evolved, and if it evolved without being useful that would make its existence all the more remarkable.

Reviewer Reports on the First Revision:

Referees' comments:

Referee #1 (Remarks to the Author):

In the revised manuscript and their rebuttal, the authors have largely addressed my previous concerns (including my misunderstanding on the starting building block). As far as I am concerned, they have mostly answered my questions about the structural origin and evolution of the reported Sierpiński triangles self-assembled by the citrate synthase (CS). As the authors admitted, "The cryo-EM density indicated that the two residues are taking part in the interaction of the 18mer interface (8R) and the hexamer interface (80F); local density of the residues is shown in Extended Data Fig. 10d+e. Both interactions are detected by Pymol's in-built cation- π and hydrogen-bond detection routines." The ascribed formation mechanism of the assemblies via the hydrogen bonding and cation- π interactions remains as structural hypotheses rather than experimental facts due to the poor imaging resolution, which is indeed slightly flawed. Since I do not have the expertise in biology to justify issues like the evolutionary meaning of the fractal structures, I prefer leaving the biological functionality justification to be made by other reviewers. From my viewpoint, the nicely assembled fractal structures via biological building blocks rather than small chemical molecules are indeed fascinating and represent a breakthrough. I therefore endorse its acceptance for publication in Nature.

Referee #2 (Remarks to the Author):

The authors have addressed my concerns, added new data, and clarified many points. Thank you!

Referee #3 (Remarks to the Author):

The authors have done a tremendous job in revising their manuscript. The realization of the additional structure of the 54mer for the H369R mutant is a most opportune addition to the manuscript.